# Benchmarking scRNA-seq copy number variation callers

Katharina T. Schmid [1], Aikaterini Symeonidi[1,2], Dmytro Hlushchenko [1], Maria L. Richter[1], Andréa E. Tijhuis[3], Floris Foijer [3] & Maria Colomé-Tatché [1,2,4] ✉

Copy number variations (CNVs), the gain or loss of genomic regions, are associated with disease, especially cancer. Single cell technologies offer new possibilities to capture within-sample heterogeneity of CNVs and identify subclones relevant for tumor progression and treatment outcome. Several computational tools have been developed to identify CNVs from scRNA-seq data. However, an independent benchmarking of them is lacking. Here, we evaluate six popular methods in their ability to correctly identify ground truth CNVs, euploid cells and subclonal structures in 21 scRNA-seq datasets. We discover dataset-specific factors influencing the performance, including dataset size, the number and type of CNVs in the sample and the choice of the reference dataset. Methods which include allelic information perform more robustly for large droplet-based datasets, but require higher runtime. Furthermore, the methods differ in their additional functionalities. We offer a benchmarking pipeline to identify the optimal method for new datasets, and improve methods' performance.

Copy number variations (CNVs) describe the gain or loss of genomic regions, from small sequences up to complete chromosomes. These genomic alterations lead to aneuploidy and are associated with different diseases and cancer types[1]. Specific CNVs are hallmarks for the classification of tumors, and are related to tumor progression and treatment outcomes[2–4]. However, the direct functional consequences of CNVs are not yet fully understood[4]. Tumors are very heterogeneous, with different tumor cells having distinct molecular phenotypes. This also applies to CNVs, which can differ substantially between cellular subclones among samples and within the same sample, emphasizing the importance of cell-specific analyses[5–7]. Single-cell whole-genome sequencing is considered the gold-standard technique to obtain per-cell CNV profiles[8], as changes in DNA copy number should lead to observable changes in the read sequencing depth. However, the technology is not frequently used in the laboratory compared to other single-cell technologies.

Instead, computational methods have been developed to infer the CNV profiles from single-cell RNA-seq data[5,9–14] and single-cell assay for transposase-accessible chromatin with sequencing (scATAC-seq) data[15,16]. These approaches have the advantage that, apart from the copy number gains and losses, the information about the cellular state can also be obtained from the same measurement (gene expression or open chromatin). For scATAC-seq, the read-out is relatively similar to whole-genome sequencing, as the genome is also sequenced, and therefore, the read coverage provides information about ploidy. However, for scRNA-seq, the inference of CNVs is challenging, as the expression level of genes is highly affected by regulatory mechanisms, and therefore, it provides only indirect information about the CNV state. Nevertheless, the general assumption of all computational methods that infer CNVs from scRNA-seq data is that genes in gained regions show higher expression, and in lost regions lower expression, compared to genes in

[1]Biomedical Center (BMC), Physiological Chemistry, Faculty of Medicine, LMU Munich, Munich, Planegg-Martinsried, Germany. [2]Institute of Computational Biology, Computational Health Center, Helmholtz Zentrum München, German Research Center for Environmental Health, Neuherberg, Germany. [3]European Research Institute for the Biology of Ageing, University of Groningen, University Medical Center Groningen, Groningen, The Netherlands. [4]Hospital del Mar Research Institute (HMRIB), Barcelona, Spain. ✉e-mail: maria.colome@bmc.med.lmu.de

diploid regions. This requires all methods to have sophisticated data normalization strategies, using generally reference diploid samples, often in combination with denoising approaches, before the different CNV inference strategies can be applied.

Because of the wealth of scRNA-seq data available, the correct identification of CNVs from this data modality is crucial to studying the role of CNVs in cancer and other aneuploid tissues. scRNA-seq CNV callers are currently used in many applications, e.g., ref. 17–20. However, there is no independent validation that shows whether scRNA-seq CNV calling methods can correctly identify CNVs, and which of the CNV callers works the best.

In this work, we benchmark six popular CNV callers for scRNA-seq data using 21 different datasets. We include datasets generated with different technologies, droplet-based and plate-based, and from different organisms, such as humans and mice. We evaluate the general CNV prediction performance for each method, comparing its results to a ground truth provided by an orthogonal CNV measurement (either (single cell) whole-genome sequencing ((sc)WGS) or whole exome sequencing (WES)), using correlation, area under the curve (AUC) values and F1 scores. We also assess the prediction of CNVs on a diploid sample, the correctness of the inferred clonal structure, the impact of the selected reference dataset on the performance, and the runtime and memory requirements. In addition, we evaluate the automatic identification of cancer cells for the methods that allow it. Our evaluation is publicly available with a reproducible Snakemake pipeline (https://github.com/colomemaria/benchmark_scrnaseq_cnv_callers)[21]. This enables the direct testing of new datasets to determine optimal CNV calling strategies, and it facilitates comparisons between methods to improve the performance of newly developed computational tools.

## Results

### scRNA-seq CNV calling benchmarking

We included in our benchmarking study six CNV calling methods that were developed specifically for scRNA-seq data (Table 1). The methods can be broadly classified into two categories: one class that uses only the expression levels per gene, consisting of InferCNV[5], copyKat[12], SCEVAN[13] and CONICSmat[9]; and a second class that combines the expression values with minor allele frequency (AF) information, consisting of CaSpER[11] and Numbat[14]. CaSpER and Numbat use AFs per SNP called directly from the scRNA-seq reads, and both models implement a Hidden Markov Model (HMM) to call CNVs. Also, InferCNV identifies CNVs using an HMM, but based on expression levels only. copyKat and SCEVAN both apply a segmentation approach, while CONICSmat estimates the CNVs based on a Mixture Model. All methods were run as recommended in the respective tutorials or based on default parameters.

The output of the CNV prediction depends on the method (Table 1). Half of the methods report the results per cell (CONICSmat, copyKat and CaSpER), while InferCNV, SCEVAN and Numbat group cells into subclones with the same CNV profile. Also, the resolution differs, with CONICSmat reporting the results only per chromosome arm, and all other methods either per gene or per segment consisting of multiple genes. Several of the methods have two possible outputs: a discrete CNV prediction and a normalized expression score; in these cases, both outputs were included separately in the evaluation and were abbreviated as "(CNV)" and "(Expr)", respectively. More details can be found in the Methods.

We tested all scRNA-seq CNV callers on 21 different single cell RNA-seq datasets (Fig. 1), comprising 13 human cancer cell lines (nine gastric cell lines, two colorectal adenocarcinoma lines (COLO320, HCT116), one breast cancer line (MCF7) and one melanoma cell line (A375)), six human primary tumor samples (three acute lymphoblastic leukemia samples (iAMP21, ALL1, ALL2), two basal cell carcinoma (BCC) samples and one multiple myeloma (MM) sample), one mouse primary tumor sample and one human diploid dataset (peripheral blood mononuclear cells (PBMCs)) (Supplementary Data 1). Seventeen datasets were measured with droplet-based technologies, and the four others with a plate-based technology.

Different metrics were included for benchmarking, most of them based on the comparison with a ground truth for the CNVs. We obtained this ground truth from either (sc)WGS or WES data (Supplementary Data 1). The different datasets showed large variation in CNV distribution, with CNVs covering between 7% and 93% of the total genome, and more gained regions than lost regions in the majority of the datasets (Supplementary Fig. 1). Since the scRNA-seq methods are only able to predict the CNV status for genomic regions comprising genes, while the WGS ground truth covers (nearly) the complete genome, we could only compare modalities in gene regions. As in most cases, the ground truth was not measured in the same set of cells as the scRNA-seq, and was, in some cases, obtained from bulk measurements, we combined the per-cell results from the scRNA-seq methods to an average CNV profile, called pseudobulk, before the comparison. For the plate-based datasets, where scRNA-seq and scWGS were measured in the same cells, a cell-by-cell comparison was also performed.

We applied threshold-independent evaluation metrics using correlation and AUC scores. For the AUC scores, predictions were evaluated separately for gain versus all and loss versus all, resulting in two scores. Not the complete range of thresholds is biologically meaningful for classifying regions as gains or losses, as every method defines a baseline score. For this reason, we chose to additionally calculate a partial AUC[22,23], with a maximal sensitivity defined by the baseline score so that only thresholds up to the baseline score were evaluated for losses and only scores higher than the baseline score were evaluated for gains (see Methods, Supplementary Fig. 2). Of note, partial AUC values below 0.5 indicate that most thresholds are outside the biologically meaningful value range (see Methods).

**Table 1 | CNV calling methods from scRNA-seq**

| Method (tested version) | Model | Input | Output resolution | Explicit reference optional | Cancer cell identification |
|---|---|---|---|---|---|
| InferCNV (v1.10.0) | HMM & Bayesian MM | Expression | Gene and subclone | No | No |
| CONICSmat (v0.0.0.1) | Mixture model | Expression | Chromosome arm and cell | Yes | No |
| CaSpER (v0.2.0) | Expression HMM & BAF signal shift | Expression & Genotypes | Segment and cell | No | No |
| copyKat (v1.1.0) | integrative Bayesian segmentation | Expression | Gene and cell | Yes | Yes |
| Numbat (v1.4.0) | haplotyping AFs & combined HMM | Expression & Genotypes | Gene and subclone | (Yes) | Yes |
| SCEVAN (v1.0.1) | segmentation with a variational region growing algorithm | Expression | Segment and subclone | Yes | Yes |

*HMM* hidden Markov model, *MM* mixture mode, *(B)AF* (B-)allele frequency.

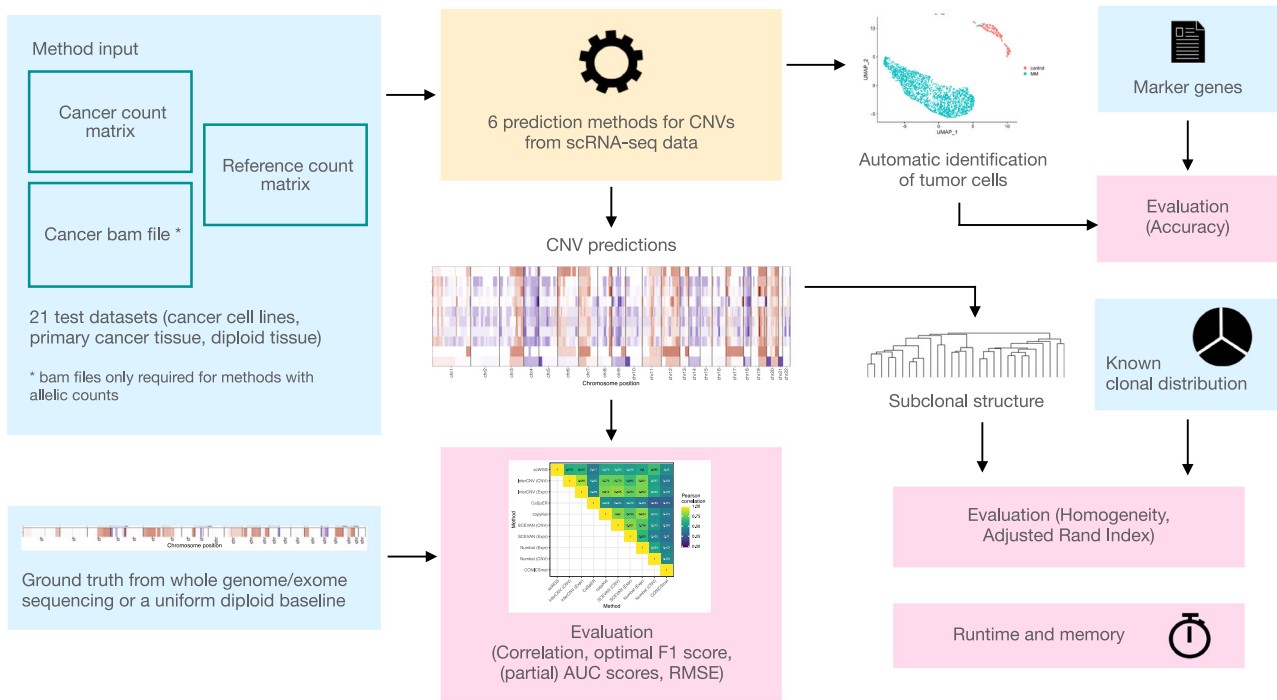

**Fig. 1 | Overview of the benchmarking workflow.** Input data in blue, evaluation results in pink.

With the same systematic, we evaluated the optimal gain and loss thresholds based on a multi-class F1 score (Supplementary Fig. 3), testing again only biologically meaningful gain and loss thresholds. These thresholds were then used to obtain sensitivity and specificity values for gains and losses.

Every scRNA-seq method requires a set of euploid reference cells to normalize the expression of the analyzed cells. For the primary tissue samples, the common assumption is that the measured tissues are a mixture of tumor and normal cells, of which the latter can be used as a reference. Some methods rely on cell type annotations provided by the user to specify the reference, while other methods provide two options: user-provided cell type annotations or automatic detection of normal cells. To ensure reproducibility between methods, cells were annotated manually into tumor and healthy cells per sample, and the same healthy cells were used as reference for all methods unless specified otherwise. We applied the published cell type annotation for the BCC and ALL datasets and performed manual annotation based on Louvain clustering and known marker genes for the MM, the iAMP21 and the mouse datasets (see Methods).

For the cancer cell lines there exist no directly matched reference cells and therefore we chose, for each dataset, a matched external reference dataset with healthy cells from the same or at least very similar cell types (Supplementary Data 1). Since the choice of the reference euploid dataset used for normalization may affect the final CNV calling results, we tested the impact of different references on the prediction quality.

Another challenge for scRNA-seq callers is the ability to detect completely euploid datasets, with no presence of CNVs. The methods' performance on an euploid dataset was not evaluated in the associated publications. However, the identification of the lack of CNVs is also an important asset. For this reason, we included an euploid dataset, comprising PBMCs from a healthy donor[24], in our performance test and calculated the mean square error deviation for every method compared to a diploid reference genome. Thereby, we explored how the performance changes depending on the choice of the reference dataset. Furthermore, for the methods with automatic detection of normal cells, we estimated the accuracy of this feature by comparing

the methods' cell assignment to the ground truth cell type obtained from the analysis of the scRNA-seq data.

All the tested methods can detect heterogeneity in the analyzed samples. CaSpER, CopyKat and CONICSmat estimate the CNV profiles per cell, which can be clustered afterwards into subclones, while Numbat, InferCNV and SCEVAN cluster the cells already during the analysis to improve the CNV prediction. To explore how well the methods can map cells to separate sub-clones with distinct CNV profiles, we mixed patient data from tumors that have been shown to display high inter-individual heterogeneity in their CNV profiles into one dataset. Running every method on this dataset, we quantified their ability to recover the different donors as different clones.

All evaluations were set-up within a Snakemake pipeline[21], so that both new methods and new datasets can be easily integrated into the benchmarking.

**Benchmarking scRNA-seq CNV prediction compared to genomic ground truth in droplet-based data**

We evaluated all CNV callers on 15 different human cancer datasets measured with droplet-based scRNA-seq technologies (Supplementary Data 1) using various metrics. On average, Numbat (Expr), copyKat and InferCNV (Expr) had the highest maximal F1 scores (between 0.59 and 0.57), and also scored high for all other metrics (Fig. 2A, Supplementary Fig. 4). However, comparing the maximal F1 scores shows that the performance differences between the callers were in most cases non-significant (Supplementary Fig. 5). More significant differences were visible for the correlation and partial AUC scores. Furthermore, all metric scores for all methods showed a large standard deviation across datasets (Fig. 2A, Supplementary Fig. 4). Due to these aspects, no method can be seen as clearly superior to the others.

The different metrics give insights into different aspects of the prediction. The maximal F1 score puts equal weight on predicting all three CNV classes (gain, base and loss), independent of their occurrence in the dataset. This is important as most of the datasets have more gains than losses, and some of the cell lines (HGC27, KATOIII and SNU16) have very extreme profiles with >75% of gained regions (Supplementary Fig. 1). A method/dataset combination with high maximal

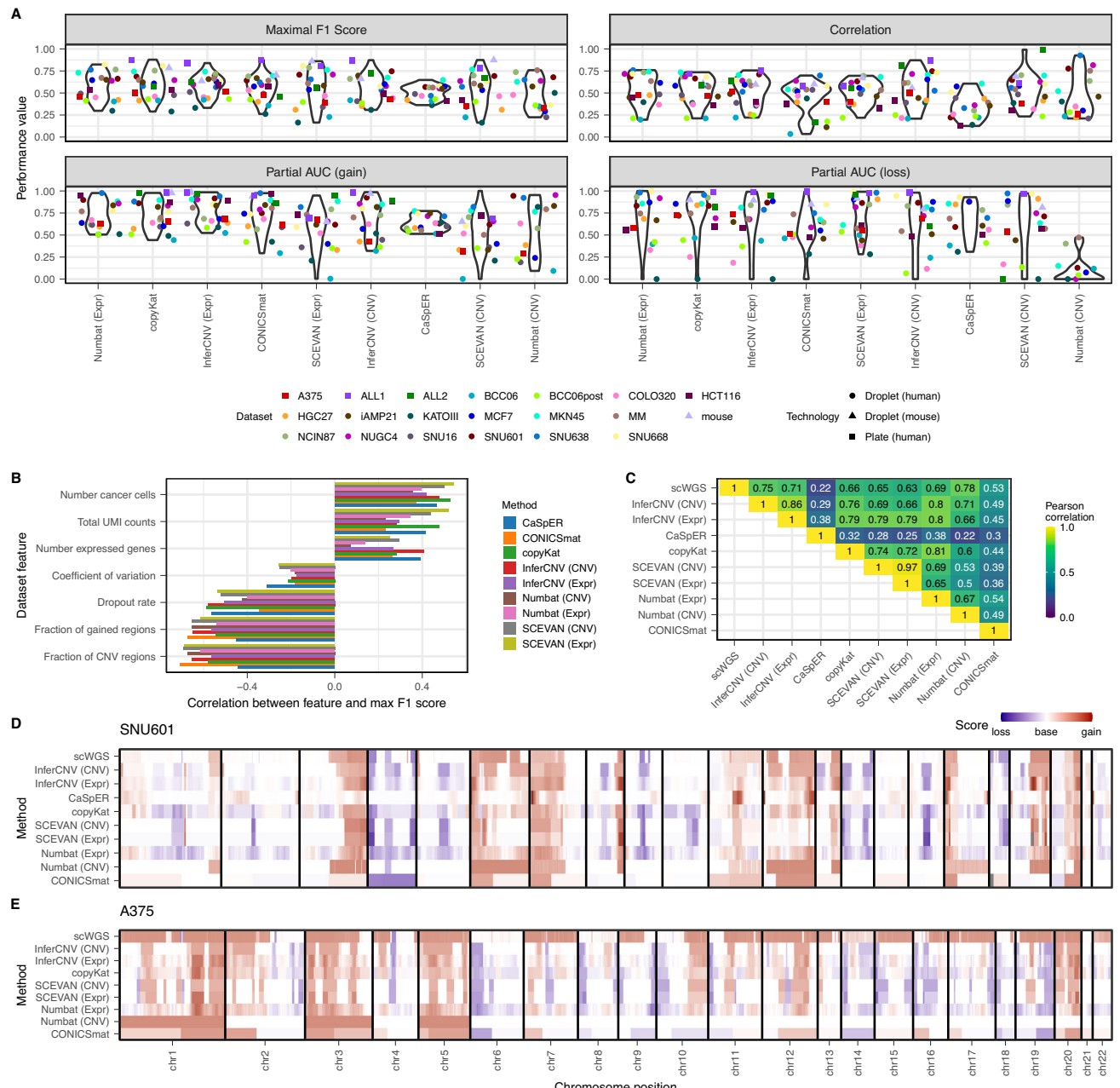

**Fig. 2 | General performance evaluation on aneuploid datasets. A** Performance comparison across all datasets. Due to a lack of genomic information, the ALL1, ALL2 and mouse data were not run with CaSpER and Numbat. For the dataset A375, CaSpER identified no CNVs, and for HCT116, Numbat (CNV) identified no CNVs. **B** Impact of dataset characteristics on the performance (maximal F1 scores). The features show the total number of UMI counts, the number of cells and the number of expressed genes in the cancer dataset, the mean dropout rate per cell, the mean coefficient of variation across genes, the fraction of gained regions from all ground truth annotated regions and the fraction of CNV regions. Illustrating the top 7 general dataset characteristics with the highest mean absolute correlation to the performance, all features are shown in the supplement (Supplementary Fig. 8). **C** Method comparison within the SNU601 dataset. **D**, **E** Karyogram of the SNU601 dataset (**D**) and the A375 dataset (**E**), every method score was scaled to the same standard deviation. Source data are provided as a Source Data file.

F1 score indicates that the method was able to predict all three classes accurately. The AUC and partial AUC scores evaluate instead the separate performance between calling gains and losses. The correlation score indicates how well the overall genome-wide profile is recovered, and may reflect only the majority call.

We see that three of the CNV callers (CONICSmat, Numbat (CNV) & InferCNV (CNV)) are better at predicting gains, as they show a significantly higher sensitivity for gains compared to losses (Wilcoxon signed rank test, FDR < 0.05; Supplementary Fig. 6). Two more methods show at least slightly significantly higher sensitivity for gains (InferCNV (Expr) & CaSpER; FDR < 0.1). For example, in the SNU638 cell

line, which contains gains but nearly no losses (Supplementary Fig. 1), Numbat (CNV) predicts the gains very well, visible in high gain sensitivity and precision (>0.9), but it does not identify any loss regions (loss sensitivity = 0) (Fig. 2A). For this reason, the maximal F1 score is only 0.65. However, the method exhibits a very high correlation (0.93) in that dataset, because despite not identifying any loss region at all, the overall profile was correctly predicted (Supplementary Fig. 4).

CNVs can be further classified, based on their length, into focal CNVs of short length and broad CNVs[25]. There are different biological mechanisms behind both types, and often focal CNVs are associated with cancer-driving genes[26]. However, the detection of focal CNVs is

difficult due to their size, prompting us to analyze specifically how well the different callers can identify them. The high resolution of the ground truth scWGS data allowed us to categorize the CNVs into focal and broad for the respective datasets. We tested three different definitions of focal amplifications, based on their size: focal type 1 (size <3 Mb)[1,26,27], focal type 2 and focal type 3 (size <50% or <95% of a chromosome arm, respectively)[25,28]. All datasets contained several focal CNVs for all three definitions; however, for focal type 1 CNVs, only a few overlapped the regions analyzed by the CNV calling methods (Supplementary Fig. 7A, B). This highlights a general coverage problem of all scRNA-seq callers for very small CNVs. Also, the sensitivity of the individual callers for this subset of focal type 1 CNVs was very low (Supplementary Fig. 7C). However, the detection sensitivity of focal type 2 & 3 CNVs was similarly high to the one for all CNVs, so we see that only focal CNVs according to the most strict definition are affected. For this subset, we conclude that all scRNA-seq callers are, in general, not suitable for identifying them.

To study the variable performance between datasets, we explored different data characteristics influencing the CNV calling (Fig. 2B, Supplementary Fig. 8). We saw a positive correlation between performance (maximal F1 score) and number of cancer cells, number of expressed genes, as well as read coverage (UMI counts). Vice versa, the performance was negatively correlated with the dropout rate and moderately with the gene expression variation between cells. The strongest negative correlation, however, was observed with the fraction of genomic aberrations: the larger the fraction of the genome in the gain state, the more difficulties all methods had in inferring the correct CNV profile. This is probably due to the fact that all methods seemed to have a problem identifying the baseline ploidy in these extreme cases (Supplementary Fig. 1, Supplementary Data 2). The same trend was also visible for the fraction of CNV regions in general, which is, however, strongly correlated with the fraction of gained regions for the tested datasets. In summary, several dataset characteristics explained the deviations in performance between datasets, and all methods were similarly affected.

For some of the CNV callers, the predictions tended to agree more between methods than when compared to the genetic ground truth (Fig. 2C, D, Supplementary Fig. 9, Supplementary Data 2). This was in particular the case for InferCNV, copyKat, SCEVAN and Numbat (Expr) (i.e., correlation of Numbat (Expr) with InferCNV, copyKat and SCEVAN was significantly higher than with the ground truth (Wilcoxon signed rank test, FDR < 0.05)). This could be caused by true CNV differences between the scRNA-seq and genetic datasets, because the cells analyzed as genetic reference were not the same ones used for the scRNA-seq analysis. It could also reflect technical and biological biases of scRNA-seq data, which were picked up by all top-performing methods similarly, such as problems with lowly expressed genes or with pathways upregulated in cancer.

The methods differed in their data quality filtering steps, which ultimately led to a different number of included genes and included cells in the CNV analysis. A more lenient gene expression filtering leads to more annotated genomic regions for CNVs. In our evaluations, we compared the CNV callers using only the overlap of all the considered regions. We therefore tested the change in performance when considering all covered regions per method, instead of only the intercept, using the SNU601 dataset, and observed no change in performance (Supplementary Fig. 10). CaSpER and CONICSmat kept the most genes (Supplementary Fig. 11). Despite calling CNVs for a larger portion of the genome, the permissive expression filter can negatively affect a method's performance. This is the case for CaSpER, one of the methods with the lowest correlation values on average and no maximal F1 score above 0.65 on any dataset (Supplementary Fig. 4). To show how the expression filtering influenced the performance, we exemplarily ran CaSpER with a more strict cutoff on the SNU601 cell line (keeping 8847 genes instead of 13,196 genes) and saw a clear improvement in the CNV calling results (Supplementary Table 1). CONICSmat might not be as affected by the lenient expression cutoff, as it provides CNV predictions per chromosome arm, while all other methods allow for a far higher resolution. In general, an extensive parameter optimization of all methods is out of the scope of this benchmarking. We ran each method with the recommended default parameters. The users should, however, be aware that methods might perform better with other parameters.

In addition to the pseudobulk evaluations, we also checked the per-cell estimates. For the SNU601 cell line, the pseudobulk CNVs were closer to the ground truth compared to the per-cell results for all methods, probably due to the reduction of noise (Supplementary Figs. 12 and 13). We saw a performance difference between the methods, which output subclone or per-cell CNVs, where the first showed smaller deviation from the pseudobulk profiles. This is expected, as the subclonal aggregation is conceptually similar to the pseudobulk aggregation, but on a smaller scale. Still, independent of the approach, the mean per cell correlation was over 0.45 for all methods except CaSpER and CONICSmat. So, the per-cell CNV profiles of the methods are reasonable for use in downstream analysis, although an aggregation to the subclonal or dataset level provides more reliable results.

## Benchmarking CNV prediction in other organisms and sequencing technologies

We extended the performance evaluation using other single-cell technologies and organisms. This includes paired methods, which enable the analysis of RNA and WGS in the same cell and thereby a per-cell comparison of the CNV prediction results between RNA and WGS. In total, we included four paired plate-based datasets, measured with DNTR-seq[29], and one mouse dataset (droplet-based) (Supplementary Data 1). In principle, all methods allow the running of data generated by plate-based RNA technologies and data from other species apart from humans. The newer CNV callers copyKat, Numbat and SCEVAN, however, were developed mainly for human data generated by droplet-based methods.

First, we analysed the CNV results for the two paired plate-based cell lines HCT116 and A375. For the expression-based methods, the performance was close to the mean performance of the droplet-based datasets (Fig. 2A, E, Supplementary Fig. 4, Supplementary Data 3 and 4). However, for the methods including AF information, CaSpER and Numbat, the performance dropped considerably, with few to no CNVs identified. This could be due to the fact that, compared to droplet-based datasets, an order of magnitude fewer SNPs could be called, likely due to the low number of cells per experiment (Supplementary Fig. 14). We analysed the other two plate-based paired datasets, ALL1 and ALL2, only using the expression-based methods, where they showed above-average performance (Fig. 2A, Supplementary Fig. 4, Supplementary Data 5 and 6).

Thanks to the paired information, we could perform a cell-by-cell comparison of CNV profiles (Supplementary Data 3–6, Supplementary Fig. 15). The HCT116 cell line and the ALL2 dataset were very homogenous based on the DNA CNVs, and the same homogeneity was visible at the RNA CNV level (Supplementary Data 3 and 6). In contrast, the A375 cell line and the ALL1 dataset showed more heterogeneous CNV profiles across cells in the DNA data (Supplementary Data 4+5). Based on CNVs called from the scWGS data, the cells of the ALL1 dataset were categorized into two subclones in the original publication[29], the larger with an additional gain in chromosome 6. All methods performed significantly better at predicting the profiles of the larger subclone (Wilcoxon rank sum test, FDR < 0.05; Supplementary Fig. 15C). Generally, the per cell performance was on average lower than the pseudobulk performance (Supplementary Fig. 15), similar to the results for the SNU601 cell line discussed previously (Supplementary Fig. 13). Especially for ALL2, it was visible again that the methods which groups

cells into subclones (InferCNV (CNV) and SCEVAN (CNV)) had estimates which were closer to the pseudobulk results.

Finally, we tested the performance of the methods on another species, a mouse droplet-based dataset from a T-cell lymphoma. Some of the CNV RNA methods are restricted by the organisms they can run on. In particular, copyKat, Numbat and SCEVAN offer only human and mouse annotations as options. All expression-based methods showed very good performance on the tested mouse data, above average compared to the human droplet dataset (Fig. 2A, Supplementary Fig. 4). Numbat requires haplotype information, which was not available for the analysed mouse dataset and therefore could not be tested here. CaSpER could not detect SNPs on this dataset.

In conclusion, we saw that the CNV prediction methods also worked for scRNA-seq plate-based technologies and non-human species. However, the methods that incorporate AF information were more restricted due to the low number of inferred SNPs in the plate-based datasets and the need for haplotype information. These methods performed worse than the expression-based ones for the plate-based datasets.

### Benchmarking CNV prediction on euploid samples

Another important criterion for CNV prediction algorithms, which is usually not explored in the original methods papers, is the correct identification of euploid datasets, which display no CNVs. To consider this, we included in our benchmarking a diploid dataset consisting of CD4 + T cells from a PBMC dataset[24], combined with four different reference datasets. Defining a matched reference dataset for the CNV prediction is a common and crucial step for all scRNA-seq CNV calling methods. It should consist of an euploid version of the same (or similar) cell type, in order to normalize the tested cells and distinguish expression changes caused by CNVs from cell type-specific expression changes.

We randomly selected 50% of CD4 + T cells for the analysis, and the four reference datasets consisted of the remaining 50% of CD4 + T cells from the same sample, CD14+ Monocytes from the same sample (exemplifying the use of another cell type for normalization) (Fig. 3A, B); as well as CD4 + T cells and CD14+ Monocytes from another dataset[30] (exemplifying the situation where normal cells of the same cell type are not captured in the sample) (Supplementary Fig. 16).

We calculated the Root Mean Square Error (RMSE) between the method scores and a diploid baseline (see Methods). When using CD4 + T cells from the same dataset as reference, all methods showed medium to good performance, visible from the karyogram plots and the low RMSE scores (Fig. 3A, C). CONICSmat performed the worst with the highest divergence from a diploid genome (RMSE = 0.49), followed by CaSpER (RMSE = 0.46). InferCNV (CNV & Expr) and copyKat found minimal CNV presence (RMSE = 0.03, 0.07 and 0.04, respectively), while Numbat (CNV) was the only method to identify a fully diploid genome (RMSE = 0).

The RMSE values rose considerably when using CD14+ Monocytes instead as a normalizing reference (Fig. 3B, C). Here, copyKat, as well as the methods that include AF information, i.e., Numbat and CaSpER, performed clearly better (RMSE < 0.5). Numbat (CNV) was again the only method able to identify a fully diploid genome (RMSE = 0). All other methods identified a large portion of the genome as non-diploid.

The RMSE values were much higher when using an external reference dataset[30], independent of whether we used CD4 + T cells or CD14+ Monocytes as a diploid reference. Again, Numbat, CaSpER and copyKat were the best-performing methods (Fig. 3C, Supplementary Fig. 16). Interestingly, Numbat (CNV) was again able to identify a fully diploid genome.

The two PBMC datasets used here were both generated with the 10X Genomics technology; however, they were produced in different versions and in different laboratories, which likely had an impact on

the CNV results. These technical factors cannot be overcome by batch integration, as raw counts are required as input for calling CNVs. Even small mapping differences can influence the CNV calling performance. To show that, for the second reference PBMC dataset, we repeated the analysis with the same data but mapped with an older version of CellRanger. This reduced the performance even more (paired T test p value = 0.0038 for the CD4 + T cells and p value = 0.0008 for the CD14+ Monocyte; see Methods) (Supplementary Fig. 17).

Our results show that, in general, most methods can identify diploid samples given that an appropriate reference dataset with the same cell type is provided. In cases where closely matching reference cells are not available or at least not easily identifiable, CNV calling methods with allelic information are a good option, as they are less affected by wrongly matched reference datasets. Numbat (CNV) consistently detected a fully diploid genome in all tested scenarios.

### Benchmarking the impact of the reference on CNV detection for aneuploid datasets

In real applications with cancer data, the identification of a diploid population of the same cell type is more challenging than for the euploid example. Within tumor microenvironments measured in primary samples, a mixture of many cell types exists. Niche cells can also carry CNVs, despite not being cancer cells, which in these cases would distort CNV calling if using them as reference[31]. In the analysis of cell lines, no naturally matched healthy cells exist, and an external reference is always required. As discussed before, when the reference euploid dataset comes from a different sample, batch integration to minimize technical variation cannot be performed before CNV calling.

To better assess the influence of the reference dataset on the CNV results for the cancer samples, we analyzed two of the cancer datasets, MM and SNU601, with different references (Fig. 3D, E, Supplementary Fig. 18). For the MM primary cancer sample, the reference cells in the default analysis were the healthy cells from the tumor microenvironment. Here, additionally, we also tested different PBMC cell types as reference (T cells, B cells and monocytes from 10X Genomics[24]) (Fig. 3D, Supplementary Fig. 18A). We challenged the methods further by including as reference a healthy gastric dataset[32] (mix of fibroblasts, immune cells (B + T cells), endothelial cells, enteroendocrine cells, Chief cells, pit mucous cells, and intestinal metaplasia) as well as gastric cancer cells, which harbor CNVs on themselves (SNU601 cells[33]). For every other tested reference dataset, the quality of the CNV calls dropped significantly compared to the original reference from the same dataset (Wilcoxon signed rank test, FDR < 0.05; see Methods).

For the SNU601 dataset, we tested a second healthy gastric dataset[34], considering three different cell groups as euploid reference (epithelial and endothelial cells, fibroblasts and smooth muscle cells, and immune cells). In general, the CNV calling performance was very similar across references (Fig. 3E, Supplementary Fig. 18B), except for SCEVAN and CONICSmat, whose performance dropped when the gastric cancer cells were normalized using fibroblasts and smooth muscle cells. Additionally, we tested normalizing the SNU601 cells with the cancer cells from the MM dataset, to again exemplify how much the CNV results diverged when normalizing by a distant cell type that also contains CNVs itself (Fig. 3E, Supplementary Fig. 18B). Here, the performance dropped only significantly for the MM dataset as reference dataset (Wilcoxon signed rank test, FDR < 0.05; see Methods).

Overall, we see that the choice of the reference dataset has an effect on the CNV detection in aneuploid samples. If cells from the same sample are available, they tend to be the best reference dataset. In case an external reference is necessary, biological and technical differences should be kept as small as possible. If a cell type that itself is harboring aneuploidies is used as a reference, the CNV predictions become less reliable.

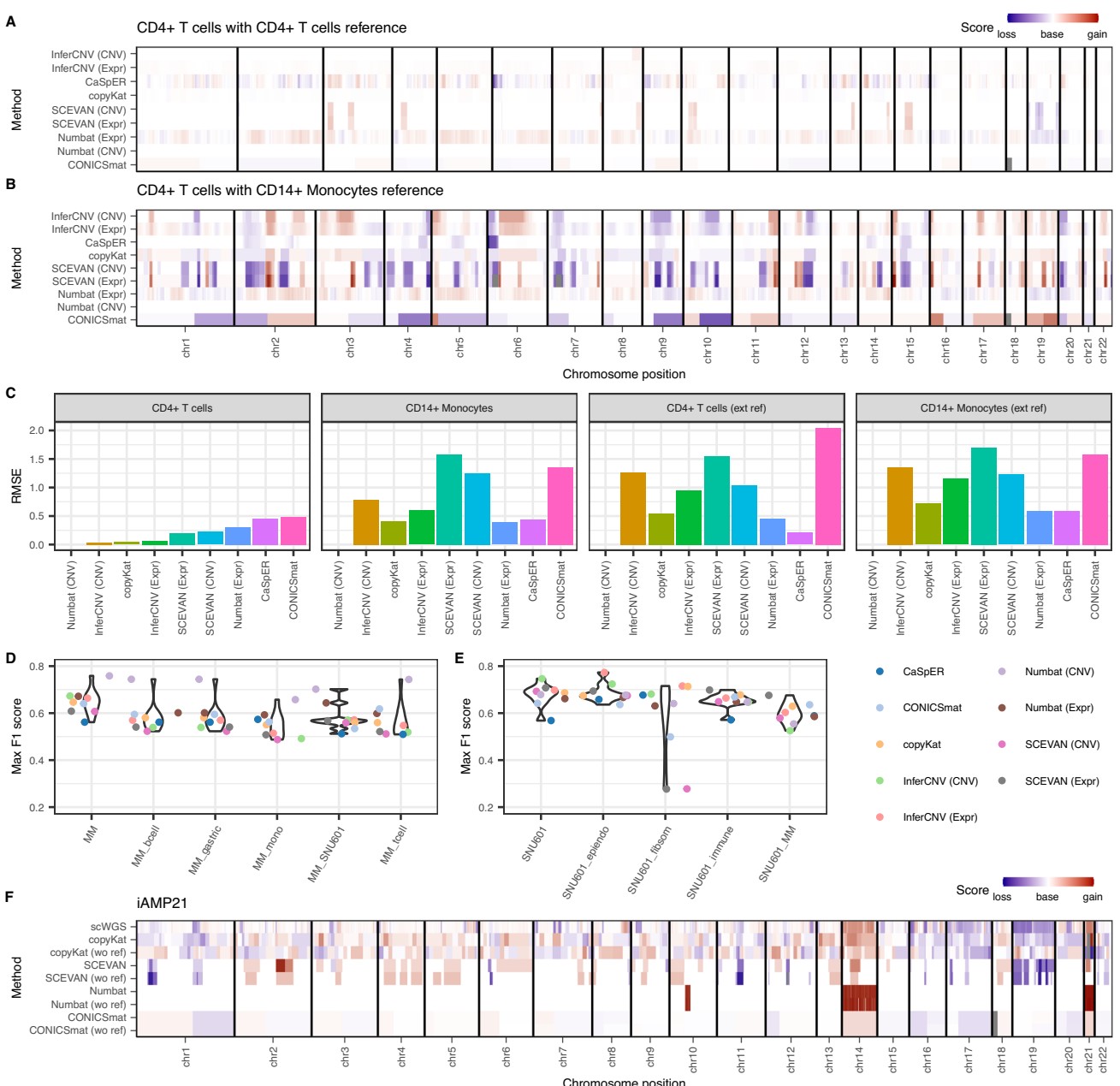

**Fig. 3 | Performance of CNV callers depending on the reference dataset for euploid and aneuploid datasets. A, B** Karyograms of CNVs in CD4 + T cells when using either CD4 + T cells (**A**) and CD14+ Monocytes (**B**) from the same dataset as reference cells. **C** Root mean squared error (RMSE) between CNV predictions of each method and diploid baseline as ground truth. The methods with the lowest RMSE perform best. Panel title shows chosen reference cells. **D, E** Performance of the methods with different reference datasets for the MM dataset (**D**) and the SNU601 dataset (**E**). For the MM dataset, we tested a second healthy PBMC dataset,

which we split into B cells, T cells and Monocytes (mono). We additionally tested a healthy gastric dataset and a gastric cancer dataset (SNU601). For the SNU601, we tested a second healthy gastric reference, which we split into three groups: epithelial and endothelial cells (epiendo), fibroblasts and smooth muscle cells (fibsom) and immune cells. We also tested the MM dataset as a reference.
**F** Karyograms showing the difference between using a manual reference and an automatic one for the dataset iAMP21. Source data are provided as a Source Data file.

## Benchmarking automatic identification of tumor cells

In order to overcome the problem of selecting the appropriate reference dataset, CopyKat, SCEVAN, CONICSmat and Numbat can be run without the input of a reference (euploid) cell set. Importantly, in the previous sections these four methods were run with an explicit annotation of reference cells, to make the results comparable between methods. Here we explored how well these four methods worked without the reference annotation. We studied the CNV calling performance changes, as well as the correct identification of aneuploid cells.

The methods use different strategies to call CNVs without the explicit annotation of reference cells. Both CopyKat and SCEVAN automatically identify putative normal cells in the dataset, which are then internally used as a reference for the CNV calling. CopyKat first clusters the cells based on gene expression and defines the cluster with the minimal estimated variance in gene expression as the diploid cells. In contrast, SCEVAN uses public gene signatures of cells from the tumor microenvironment, stromal and immune cells to identify highly confident normal cells and extends this annotation to similar cells via expression clustering; after CNV analysis, cells are clustered again

**Table 2 | Accuracy of each method in predicting the cancer cells in the respective dataset**

| | Accuracy prediction cancer cells | | | |
|---|---|---|---|---|
| Method | MM | BCC06 | BCC06 post | iAMP21 |
| CopyKat | 97.8% (99.3%) | 44.3% (44.3%) | 95.7% (95.8%) | 59.2% (59.5%) |
| SCEVAN | 97.9% (99.3%) | 97.7% (97.7%) | 95.5% (95.7%) | 27.7% (27.8%) |
| Numbat | 99.4% (99.4%) | 100% (100%) | 95.5% (95.5%) | 97.9% (97.9%) |

The first number shows total accuracy, including all cells; the number in brackets shows the accuracy when excluding cells defined as bad quality by the respective method.

based on their predicted CNV profiles for the final annotation of non-malignant cells. In contrast to CopyKat and SCEVAN, CONICSmat does not automatically report identified diploid cells. It instead uses a two-component Gaussian Mixture Model (GMM) per genomic region to identify regions with a different coverage profile as potential CNVs. Finally, Numbat uses an external reference if no internal is provided: the method contains a large gene expression reference set based on the human cell atlas and automatically matches the closest reference cell type for each cell. After running the CNV calling, the normal cells in the dataset are identified based on the aneuploidy probability, calculated from the posterior probabilities of the HMM. We tested these methods on four primary cancer datasets, MM, iAMP21, BCC06 and BCC06post (i.e., posttreatment) (Supplementary Data 1), which are supposed to be a mixture of cancer and normal cells. Their cell type annotations were taken directly from the publication for the two BCC dataset[35] or were manually annotated based on known marker genes for the MM and the iAMP21 dataset.

We first investigated the agreement between the marker-gene-based cell type annotation and the annotation of tumor and normal cells, which is provided by three of the four methods: CopyKat, SCE-VAN and Numbat. In most cases, all methods showed very high concordance when predicting tumor vs normal cells (Table 2). Numbat reached values over 95% accuracy for all four scenarios. SCEVAN performed well in all scenarios except for the iAMP21 dataset, where it classified many tumor cells as normal. This could be due to an incorrect initial identification of confidential normal cells, which is based on a default marker gene list. SCEVAN also provides the option to include other marker genes for this analysis.

CopyKat performed very well on the MM and BCC06post datasets, but had problems with the BCC06 and the iAMP21 sample, as it classified about half of the manually annotated tumor cells as diploid (accuracy of 44.3% and 59.2%, respectively). According to the manual cell type annotation, the BCC06 and the iAMP21 datasets have a very low percentage of diploid cells (2% and 6%) compared to the other datasets (16% and 47%). Probably for this reason, CopyKat does not work in that scenario. CopyKat identifies diploid cells as the cluster of cells with the smallest gene expression variation, which can be incorrect when diploid cells do not build a large cluster. In contrast, SCEVAN includes tumor markers and Numbat CNV profiles, which both give additional evidence for the identification of normal cells.

We then explored how much the actual CNV predictions differed when running the methods without providing the cancer cell annotation manually (Fig. 3F, Supplementary Figs. 19 and 20). For the MM and BCC06post datasets, because the automatic identification of cancer/normal cells performed well (Table 2) the differences were very small between the different modes for all methods (Supplementary Figs. 19A, B and 20). For the BCC06 dataset, however, copyKat underperformed when the reference was not provided (Supplementary Figs. 19C and 20), most likely due to the wrong identification of normal cells (Table 2). The same is visible for the iAMP21 dataset for copyKat and SCEVAN (Fig. 3F, Supplementary Fig. 20), again most likely caused by wrong cell annotations (Table 2). Surprisingly, the CNV results for SCEVAN in the BCC06 data improved when no reference was provided.

Overall, these results show that the automatic identification of reference cells is a good approach to overcome the additional effort of choosing a correct diploid reference, as the predicted results were very similar to the manual annotations overall. However, when the number of diploid cells in the sample is low, this strategy may fail, especially for CopyKat. On the other hand, the marker gene-based approach of SCEVAN might not work equally well for each tumor type. Finally, this approach works only when normal cells are found in the dataset, which is not always the case, especially on cell lines.

## Benchmarking the identification of subclones

The big advantage of single cell data for CNV calling, compared to traditional approaches, is that subclonal structures, i.e. tumor heterogeneity, can be identified. Each of the tested methods provide information about the subclonal structure identified based on the predicted CNVs. Some methods directly define clones (Numbat, InferCNV and SCEVAN), while others produce a dendrogram (CopyKat, CONICSmat and CaSpER) that can be split to determine clones.

Obtaining independent ground truth data about the clonal structure within a cancer sample is challenging. To overcome this difficulty, we artificially created a dataset with known clonal structure by merging different cancer samples belonging to the same cancer type, but coming from different patients with different CNV profiles. To do that, we used four BCC samples[35]. CNV analysis of every patient separately shows that different patients have different CNV profiles (Supplementary Fig. 21). Therefore, we can evaluate how effective the different methods are at sorting the patients into different CNV clusters.

All the algorithms identified several subclones in this mixed dataset (Fig. 4A). We extracted one pseudobulk CNV profile per method and clone. The discovered pseudobulk clones clustered based on the donor rather than on the used method, suggesting that overall the methods were able to distinguish between donors.

To quantify the performance of every method, we calculated Adjusted Rand Indices (ARI) and Homogeneity Scores for the identified clusters compared to the true donor composition. CopyKat, InferCNV, SCEVAN and Numbat all obtained Homogeneity Scores close to 1, meaning that most clones contained mostly cells from one donor alone. All four methods had lower ARI values (Fig. 4B), probably caused by the fact that they identified multiple CNV clusters inside each donor, which is to be expected given the individual CNV results per donor (Supplementary Fig. 21). However, SCEVAN had an ARI of 0. Despite providing a cell-type annotation file as input, SCEVAN identified only the cells from one donor (BCC08) as tumor cells (clusters scevan_1, 2, 3, and 4) and all the other donor cells were classified as normal (in this case, no CNV profile was outputted for them). CONICSmat and CaSpER had lower Homogeneity Scores, as they grouped a large number of cells from all four donors together into one big clone (clone conicsmat_1 and casper_1). All methods had some problems distinguishing between donors BCC05 and BCC07, which could be caused by relatively similar CNV profiles of both donors (Supplementary Fig. 21).

Overall, we see that CopyKat, InferCNV and Numbat are the best-performing methods in terms of clonal structure identification, as all three were able to separate donors based on their CNV profiles.

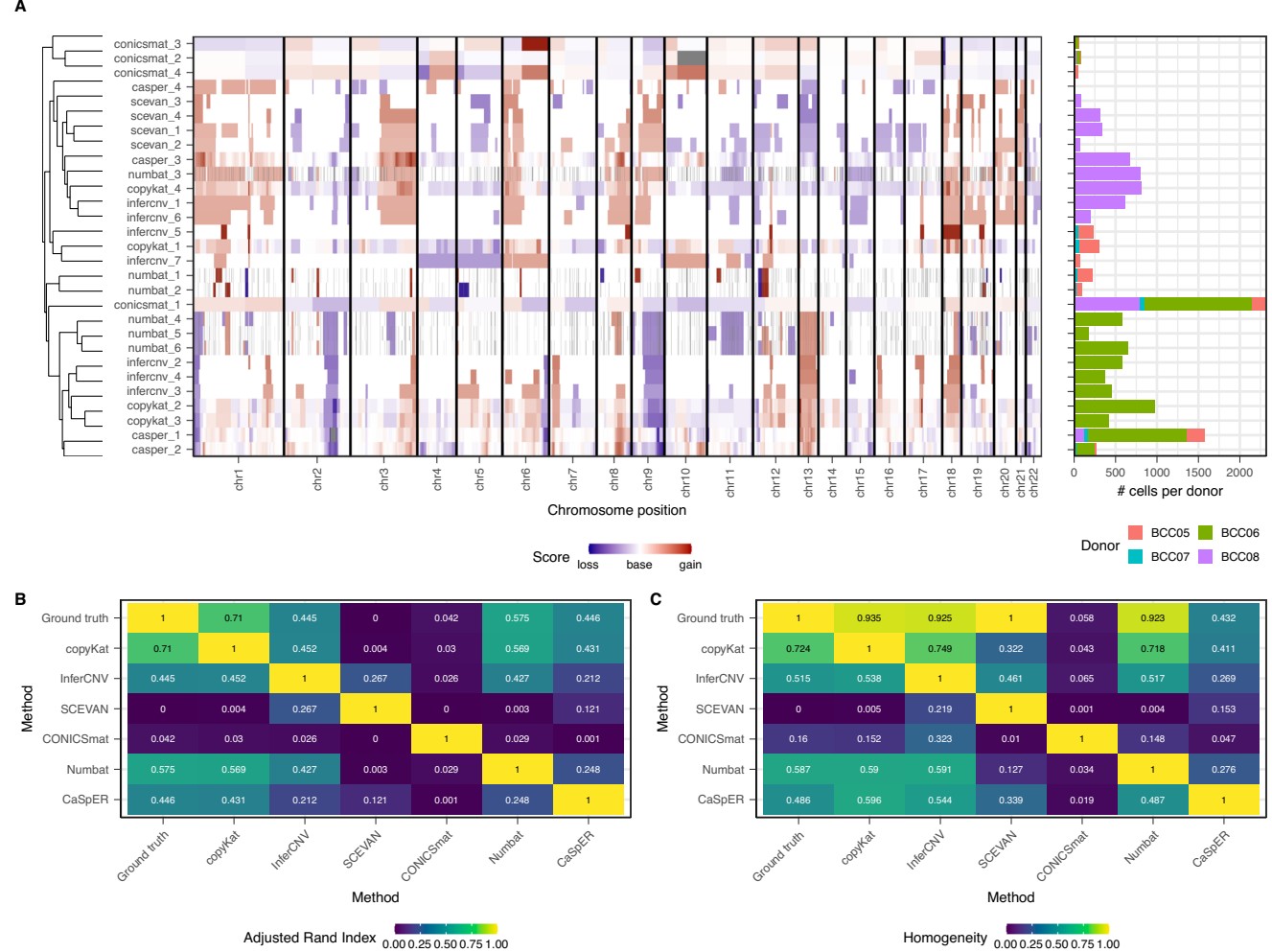

**Fig. 4 | Performance on subclone identification. A** Clustering of subclone CNV profiles for all methods. The number of cells per donor is visualized in the barplot on the right. **B** Adjusted Rand Index and **C** Homogeneity scores between the true clonal structure and the clonal structure identified by each method. Source data are provided as a Source Data file.

## Overall comparison of the methods

In summary, the methods differed more in their additional functionalities than in their ability to predict CNVs (Fig. 5), as the performance metrics for the CNV predictions on the cancer datasets showed only minor differences. The characteristics of the dataset, with enough coverage and a not too extreme CNV profile, were far more relevant than the choice of the exact CNV caller for the droplet-based human datasets. For the plate-based human datasets, all expression-based methods performed well; only the AF-based methods Numbat (CNV) and CaSpER had performance problems, likely because they found fewer SNPs compared to the droplet-based datasets. Also on the mouse data, all four expression-based methods performed well, while Numbat and CaSpER could not be run due to lack of haplotype/SNP information. Differences between methods were visible in the analysis of euploid samples. Numbat and CaSpER were better at identifying completely diploid datasets, even when choosing a reference sample that was not closely matching the right cell type.

The automatic annotation of tumor cells worked very well for Numbat in all tested datasets, while for copyKat and SCEVAN the performance was variable depending on the dataset. This additional functionality increases the usability of the methods. When testing the identification of CNV clones, the methods were able to assign most of the cells correctly except for CaSpER, CONICSmat and SCEVAN which did not succeed. In the case of SCEVAN, it did not distinguish any of the

donors as different clones, potentially as the tested dataset was very heterogeneous.

As the performance differences were relatively small in many tested scenarios, and the dataset sizes are constantly increasing, the runtime and memory requirements of the methods should also be taken into account. We observed that the methods differed quite substantially in their resource requirements (Fig. 5, Supplementary Fig. 22). CONICSmat and CopyKat were the most efficient methods in terms of runtime and memory requirements. From these, CopyKat also showed good performance results in all categories, making it a good choice combining prediction accuracy and resource efficiency. Also SCEVAN was quite fast and memory efficient, especially given that the standard analysis already includes several downstream analyses such as gene enrichments. On the other hand, InferCNV, CaSpER and Numbat had the longest runtimes. InferCNV is the oldest method and was probably designed with smaller datasets in mind. CaSpER and Numbat include AF information in their analysis; their increased robustness, shown in the detection of diploid cells, comes at the cost of more runtime. We ran each method with only one thread for better comparison, but all methods except CaSpER and CONICSmat offer options for multi-threading. Finally, CaSpER and InferCNV required the most memory, which can become a problem when running datasets with many cells. As dataset sizes grow overall, the use of certain methods will become infeasible in these cases without a large

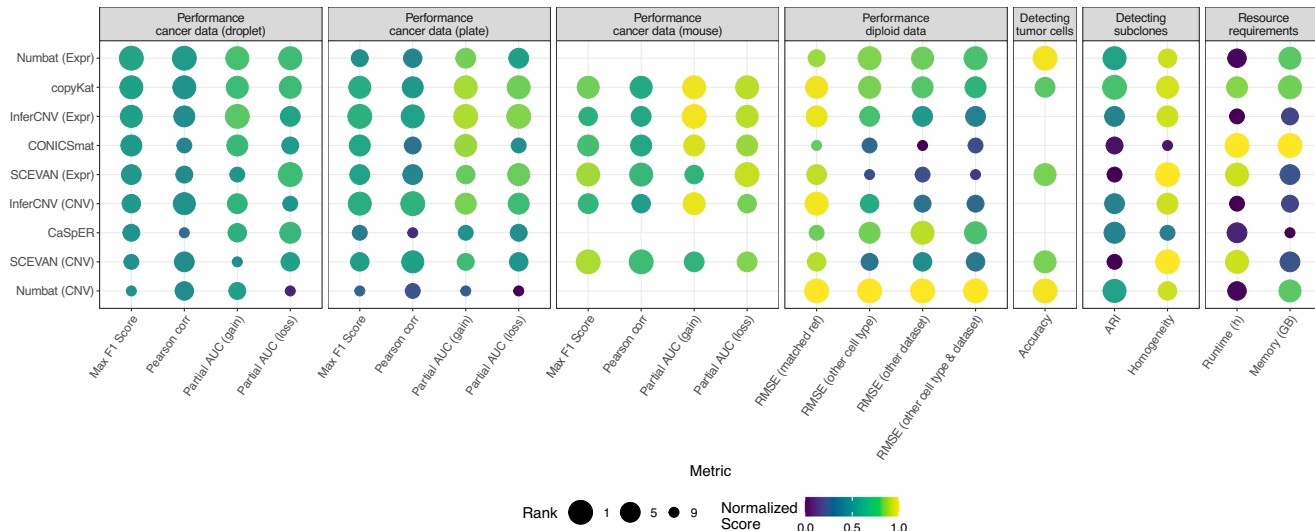

**Fig. 5 | Summary of the benchmarking results.** Main categories: mean CNV prediction performance for the different cancer datasets (human droplet-based, human plate-based and mouse droplet-based) and diploid datasets, tumor cell classification, subclonal identification and required resources. The dot size shows the rank of each method for the respective column, with 1 being the best-performing method in the category. The dot color represents a normalized score, where the values of each metric are scaled from 0 to 1, in a way that 1 is always the best value. For this, the values of RMSE, runtime and memory consumption were subtracted from one. Source data are provided as a Source Data file.

computational infrastructure. Efficiently implemented methods, like CONICSmat, CopyKat and Numbat, will clearly be an advantage here.

## Discussion

This study provides a comprehensive and independent benchmarking for scRNA-seq CNV callers. We used 21 different datasets, comprising cell lines, primary cancer tissues and a diploid dataset, and extensively evaluated six methods in their ability to recover the ground truth CNV profiles, their capacity to identify the lack of CNVs in diploid samples, the correct identification of tumor vs healthy cells, and the accurate identification of subclonal CNV structure in the data. We carefully selected and designed a set of metrics, as the CNV prediction is a three-class problem with continuous scores, but a restricted, biologically meaningful value range. We applied either threshold independent evaluation metrics, such as the correlation and (partial) AUC scores, or chose the thresholds that produced the maximal F1 scores.

The benchmarking did not identify one single method that outperformed all others in all tasks. While Numbat (Expr), copyKat and InferCNV (Expr) outperformed at predicting CNVs in aneuploid datasets from droplet-based methods, the expression-only methods were better suited for plate-based methods. Numbat (CNV) outperformed all other methods at detecting the right CNV profile in euploid samples regardless of the reference diploid dataset used. The adequateness of the cell type used as a diploid reference had a strong effect on the correct identification of diploid cells for all other methods. Furthermore, using cells from an external reference reduced the performance, even when using a matching cell type. This is to be kept in mind when designing experiments, if the reference euploid cells are measured separately from the aneuploid ones. The effect of the best matching reference sample was also visible on the CNV discovery in aneuploid samples, where again reference cells from the same dataset gave the best performance.

Dataset quality had far more impact on the CNV results than the chosen algorithmic solution. In particular, both the number of cancer cells and the total UMI counts in the dataset positively influenced the performance of all methods, while the dropout rate, the amount of gene expression variation between cells, and the fraction of the genome in the gain state negatively influenced it. For several methods, losses were more difficult to identify compared to gains. Specifically

for the sparse single-cell data, differentiation between dropouts and low coverage due to copy number losses is challenging. Furthermore, previous studies about CNVs reported that losses occur more frequently in genomic regions of low gene density[28], another disadvantage for scRNA-seq callers. Also, the size of the CNV has an effect on the detection sensitivity, as short focal CNVs were more difficult to detect.

The methods had very different performances in their other functionalities. Three of the evaluated methods, copyKat, SCEVAN and Numbat, are able to identify tumor vs healthy cells. Numbat outperformed all other methods at this task. When only a few normal cells were present in the sample, copyKat wrongly identified some of the cancer cells as normal. SCEVAN did not work in the leukemia dataset, likely because the marker genes used by SCEVAN for cell type identification were not suitable for identifying normal cell types in this dataset. Finally, all methods provide information about the CNV clones present in the analyzed sample. CopyKat, InferCNV and Numbat were able to identify the right clonality, while on the opposite side of the spectrum, SCEVAN was not able to disentangle any clones. In general, CNV clones called from scRNA-seq data could also be caused by regulatory changes of gene clusters that lead to expression changes, such as cell cycle effects, and need to be evaluated carefully.

In terms of resources, CONICSmat was the most resource-efficient method (both runtime and memory), but its CNV detection performance showed weaknesses and was highly influenced by the choice of reference cells. Moreover, the method could not correctly disentangle the clones present in a mixed dataset. CopKat and SCEVAN were fast methods. CopyKat's performance was high in all tests, except for the automatic identification of healthy cells if these were not numerous in the sample. SCEVAN instead had variable performance at the CNV identification tests and could not identify clones in a mixed sample, but it performed well at the identification of healthy vs tumor cells in three of four cases. InferCNV took the longest runtime and used the second most memory of all tested methods, but performed relatively well in all evaluated categories. CaSpER and Numbat were runtime-intensive methods, probably due to the incorporation of AF information. Between the two, Numbat offers more additional functionalities and scored better overall than CaSpER in all of our tests.

We focused our study on the basic distinction between gain, base and loss CNV classes to homogenize results between methods. However, some methods allow more detailed differentiation of the CNV states. For example, InferCNV reports the exact ploidy up to three gained copies. Both Numbat and CaSpER report additionally the copy-neutral loss of heterozygosity, which is possible only as they include AF information. Depending on the exact use case, these additional functionalities might be helpful. Methods including AF information cannot be applied to data from mouse strains, which carry too little heterozygous variation.

We included both primary cancer samples and cell lines in the benchmarking, which differ slightly in their analysis, as cell lines require external reference datasets. This issue applies not only to the commercial cell lines we used in our test cases, but also to any newly established cell line in the laboratory. This makes our results, specifically for analyzing cell lines, relevant for different scenarios. We see, however, no significant differences in method performance between the cell lines and the primary cancer samples (Wilcoxon rank sum test, FDR > 0.05).

The usability of the tools was not evaluated, as an objective quantitative metric for this is difficult to obtain. All analyzed methods provided a tutorial with installation instructions and an example analysis. CopyKat, Numbat and SCEVAN can only be run for a predefined set of gene annotations; therefore, they can only be applied to human and mouse datasets on a limited set of genome versions. Some of the methods, i.e., Numbat, SCEVAN and InferCNV, report discrete CNV classes; all other methods output a continuous CNV score, which is not straightforward to discretize and interpret by the users. Moreover, methods differ largely in their downstream functions, which were not benchmarked here. As several of the methods were developed recently, an active user community can potentially help with their feedback to overcome current limitations. We tested but did not include two additional CNV calling methods, HoneyBadger[10] and sciCNV[36]. HoneyBadger did not terminate running on the SNU601 dataset (5880 cells) after multiple days, likely because it was not developed for droplet-based datasets with many cells. For sciCNV, there were errors in the code, which prevented us from executing the method's tutorial.

There is a general debate whether CNV profiles can be accurately inferred from scRNA-seq data, as the gene expression is only indirectly associated with the CNV profile and additionally influenced by other regulatory factors, and RNA-seq data covers only a small part of the genome. Nevertheless, as it is the most abundant single-cell technology, the prediction of CNVs from it represents an appealing possibility. In our benchmarking, we showed that all tested RNA CNV callers were able to infer the CNV profile with relative accuracy, given that the analyzed dataset had sufficient coverage and a not too extreme CNV profile.

We have previously shown that the prediction of CNVs from other omics layers, such as scATAC-seq, performs better in comparison to scRNA-seq[16]. The establishment of multi-omics methods offers promising new opportunities to improve CNV calling in the future[37]. Furthermore, CNV calling methods are currently being extended to spatial transcriptomics data[38,39]. Our benchmarking not only provides a guideline for the analysis of scRNA-seq CNVs but also represents the basis for developing and improving CNV calling methods. To support the community, we have provided a reproducible Snakemake pipeline[21], such that users can test their datasets against all implemented methods, and developers can test the performance of their new methods against a wide range of datasets and functionalities.

## Methods
### Ethical statement
All animal protocols were approved by the Central Committee for Animal experiments (CCD; permit AVD10500202215846) and UMCG Committee for Animal Care (IvD).

## Running the CNV RNA-seq methods
We ran each method on the droplet-based datasets with their default parameters or the parameters suggested for droplet-based datasets in their tutorials. The human datasets were run on annotations based on hg38, the mouse dataset based on mm10; otherwise, all other parameters were kept the same. For the DNTR-seq datasets[29], we followed the recommendations for Smart-seq data, using the raw read matrix as input unless described otherwise for the specific method. Runtime and memory consumption of each method were automatically tracked within the snakemake workflow[40] (v7.32.4), which was used to apply each method to all datasets.

Details about each method are described in the following, including changes implemented for the DNTR-seq datasets:

### InferCNV
InferCNV identifies CNVs based on an HMM model, which is followed by a Bayesian Mixture Model to remove potential false positive predictions. It is available as an R-package on BioConductor, we evaluated version 1.10.0. We ran the default six-state HMM model with the analysis mode subclusters to identify potential different CNV states between subclones. The default analysis mode is sample, but the documentation states that the subcluster mode is slower, but more accurate, so we chose this. Gene annotations were taken from the tutorial website, based on Gencode hg38 annotations. For the droplet-based datasets, the filtering parameter cutoff was set to 0.1, which represents the minimal average count threshold per gene among the reference cells. This value was recommended for 10X data in the tutorial. All other parameters were kept to their default values. For the DNTR-seq datasets, we changed the expression cutoff from 0.1 to 1.

The final results comprise two different outcomes: a matrix with relative expression intensities and a matrix with pruned CNV predictions in 6 states. Both outputs are evaluated independently as InferCNV (Expr) and InferCNV (CNV), respectively. The 6 outcome states are grouped into gain - base - loss with states 1 and 2 being loss, 3 being base, and 4 to 6 being gain.

### CONICSmat
CONICSmat is an adaptation of the tool CONICS, which uses DNA sequencing data in combination with scRNA-seq data to analyze CNVs. In contrast to CONICS, CONICSmat requires no genomic data. Therefore, we included only CONICSmat in our analysis, as the inputs are comparable to those of the other tools.

CNVs are estimated based on a two-component GMM of the expression data of each cell and region. The regions can either be predefined as potential copy number alterations based on genomic data or, as in our case, without any additional data, each chromosome arm is defined as one region. CONICSmat requires a dataset with euploid and aneuploid cells to fit the two-component GMMs, but an annotation of euploid cells is optional. The tool can also be run without providing an explicit reference. We tested both options, once with and once without defining an explicit reference. Regions where the two-component GMM is statistically more likely than a uniform distribution are defined as CNV regions, and their type (gain or loss) is assigned based on the expression levels. We tested the R package version v0.0.0.1 in our benchmarking.

### CaSpER
CaSpER combines the expression signal with loss of heterozygosity information, based on SNPs called from the single-cell data. It performs multi-scale smoothing of the expression signal, followed by a 5-state HMM model for each scale. The outcome is combined with a multi-scale smoothed beta-allele frequency (BAF) shift signal to confirm the intermediate HMM CNV states. CNV calls are obtained for each combination of expression HMM and BAF shift signal. We apply 3 different scales for both expression and BAF signal, resulting in 9 CNV

calls for each segment. Following the tutorial, summary gains and losses are defined when they are called at least 7 of 9 possible times.

Preprocessing of the datasets was done with Seurat analogously to the tutorial; the cells were normalized to 1 million counts per cell and logarithmized before running CaSpER. We chose an expression threshold of 0.1 for the droplet-based datasets (parameter expr.cut-off), as applied in the 10X specific tutorial of CaSpER. For DNTR-seq data, we merged the individual bam files per cell to one combined file, used the TPM matrix (transcripts per million) as the input matrix and increased the expression cutoff to 4.5. We tested version 0.2.0 of the R package.

As an additional remark, the complete CaSpER object, generated by the package, can become very large. To save memory, we stopped storing this object as an RDS file in the end.

## copyKat

CopyKat attempts to overcome the issue of manually defining reference cells by automatic annotation of normal cells in mixed samples from tumor microenvironments. Normal cells are labeled based on the assumption that they cluster together and display the lowest variance compared to tumor cells. The CNV profile is identified via an integrative Bayesian segmentation approach.

We tested version 1.1.0, using the default parameters for filtering and segmentation. The gene position file is provided internally, but only for human (hg20) and mouse (mm10). The method was evaluated twice, once with specified reference cells and once without, so that the reference cells were identified automatically. The output file contains relative copy number ratios (instead of discrete CNV states).

## Numbat

Numbat aims to improve CNV detection by combining expression and genotype information. In contrast to CaSpER, it uses population-based phasing to get haplotype information. This is more sensitive compared to using SNPs directly, as it can overcome the sparsity to a certain extent. Expression and haplotype frequencies are combined in an HMM model. The method also infers the phylogenetic tree during the CNV prediction, so that information from cells belonging to the same subclone can be aggregated to pseudobulk, again to reduce the sparsity. For this, it alternates between CNV prediction and reconstructing the phylogenetic tree based on the identified CNVs. In the end, tumor and normal cells can be classified based on posterior probabilities for the baseline CNV.

Numbat provides a large external reference dataset for the analysis with many cell types from the Human Cell Atlas. However, we generated our own reference set per dataset, so that the same reference is used between all methods and the comparability is increased. For the DNTR-seq datasetes, the SNP calling with the respective script pileup_and_phase.R was run with the option --smartseq, the gamma parameter (the dispersion in the allele model) was changed from 20 to 5, and the min_cells parameter was reduced to 10 instead of 50. We tested the R package version 1.4.0. To check how much the AF information adds to the CNV detection, we included the normalized expression profiles of the single cells as Numbat (Expr), which does not contain any genotype information, and the clone-level pseudobulk CNV profiles as Numbat (CNV), which contains both expression and genotype information.

## SCEVAN

Similar to CopyKat, SCEVAN offers the option to automatically identify normal cells as a reference, however, with a different approach. A set of so-called confidential non-malignant cells is identified based on gene profiles of malignant and non-malignant cells, and then this set is extended to similar cells based on expression clustering. With this reference, genes are normalized and smoothed before a joint segmentation across all cells together identifies common breakpoints.

Cells are clustered afterwards to identify both healthy diploid cells and then different subclones. Afterwards, one CNV profile is obtained per subclone.

SCEVAN includes in its workflow several further analyses, such as pathway enrichment analysis, which can facilitate the interpretation of CNV results, but are not included in our benchmarking. We tested version 1.0.1 of the R package. We evaluated the normalized expression per cell, aggregated to pseudobulk, in SCEVAN (Expr) and the final CNV profile per subclone, again aggregated to pseudobulk, in SCEVAN (CNV). The final CNV prediction was simplified to 0 and 1, both being loss, 2 being base and 3 and 4 being gain. The method was evaluated twice, once with specified reference cells and once without, so that the reference cells were identified automatically.

**Experimental procedure to generate the mouse dataset.** T-ALL tumour cells were isolated from the thymi of two adult mice with mixed genetic background, one male mouse with a Mad2 f/f; p53 f/f; Lck-Cre+ genotype (T989)[41] and one female mouse with Msh2 −/− genotype (eT)[42]. The mice were kept on a day/night rhythm of 12 hours light and 12 hours dark. Humidity was controlled to be between 45 and 65% and the temperature between 20 and 24 °C with a setpoint of 21.5 °C. Sex was not considered in the downstream analyses, as it is not relevant for CNV calling in autosomal chromosomes.

The thymi were dissociated into single cells and frozen until further processing. Single-cell RNA sequencing library preparation was performed using the 10x Chromium Next GEM Single Cell 3' protocol v3.1 (eT) and the Seekgene SeekOne DD Single Cell 3' Transcriptome protocol (T989). Libraries were sequenced on a NextSeq 500 (eT; Illumina; up to 70 cycles - paired end) and NextSeq 2000 (T989; Illumina; up to 100 cycles - paired end). Sequencing data was aligned using STAR[43] (v2.7.11a).

For single-cell whole-genome sequencing, cells were suspended in media, washed, and pelleted. To generate nuclei, cells were resuspended in cell lysis buffer (100 mM Tris-HCl pH 7.4, 154 mM NaCl, 1 mM CaCl2, 500 µM MgCl2, 0.2% BSA, 0.1% NP-40, 10 µg/mL Hoechst 33358, 2 µg/mL propidium iodide in ultra-pure water) and incubated on ice in the dark for 15 minutes to complete lysis. Resulting cell nuclei were gated for G1 phase (as determined by Hoechst and propidium iodide staining) and sorted into wells of 384-well plates on a MoFlo Astrios cell sorter (Beckman Coulter). For single-cell sequencing, a single nucleus was deposited per well. Library preparation was done at the ERIBA Research Sequencing Facility.

Libraries were sequenced on a NextSeq 2000 machine (Illumina; up to 77 cycles – single end), and aligned to the murine reference genome (GRCm38/mm10) using Bowtie2[44] (v2.2.4 and v2.3.4.1). Duplicate reads were marked with BamUtil[45] (v1.0.3) or Samtools[46] markdup (v1.9).

## Preprocessing the single-cell datasets

The required input files for our pipeline are a count matrix with cell type annotations, which defines, most importantly, the cancer cells, and a BAM file to estimate AFs for Numbat and CaSpER. Bam files and count matrices were generated from the fastq files by running Cell-Ranger (v7.0.0), except for the BCC dataset, where the annotated count matrix was taken directly from GEO (based on CellRanger v2.1.0). For iAMP21, we obtained bam files, which we first converted back to fastq files using bamtofastq from CellRanger (v7.0.0).

The cell type annotation is only relevant for the primary cancer samples, as the cell lines are assumed to be a homogenous set of cells and all labeled as the cell line. For the BCC dataset, the published cell type annotation was used. For the MM dataset, a manual annotation was performed, using Louvain clustering and known marker genes for MM (CD38, IGHM, MZB1), after performing some standard filtering steps (removing cells with low counts or high mitochondrial fraction). In the same way, a manual cell type annotation of the iAMP21 dataset

was done with the following key annotation marker genes: MS4A1 for B cells, HBD for erythroid cells, CD3D for T and NK cells and TNFRSF17 for the cancer cells.

The aneuploid T cell lymphoma from the mouse (T989) was annotated the same way. It contained mostly T cells (identified using Cd3e as a marker) and a small fraction of B cells (using Cd74 & Itgax as markers). We removed the B cells, so that the dataset contained only aneuploid T cells. As a reference for this dataset, the count matrix of the second euploid T cell lymphoma (eT) was used, again after basic quality control.

In contrast, the cell lines required an external reference dataset. We selected them to match the original tissue of the cell line, i.e., intestine for SNU601, colon for COLO320 and breast for MCF7 (see Supplementary Data 1). We furthermore matched the technology, using a droplet-based reference for a droplet-based cancer dataset and a plate-based reference for a plate-based cancer dataset.

The DNTR-seq cell lines (HCT116 and A375) were downloaded in raw form and had to be processed, following the pipeline of the publication[29]. Briefly, the raw fastq files were trimmed using prinseq-lite[47] (v0.20.4), in a first round, trimming the first 10 bases of each read, low quality bases, and low complexity reads. A second round of trimming removed orphan reads, and for the removal of adapters that followed trim-gallore[48] (v0.6.11) was employed. After the reads were cleaned, they were aligned to the hg38 genome, using STAR[43] (v2.7.1a), with parameters as proposed by the authors. Following the alignment, the reads were filtered for duplicate reads, using the MarkDuplicates software from the Picard tools[49] (v3.0.0). The count matrix was generated using HTseq[50] (v2.0.5). To run CaSpER on the datasets, the individual bam files per cell were merged into one using bamtools[51] (v2.5.1).

The two primary DNTR-seq ALL samples were obtained already processed in the form of count matrices, including a cell type annotation.

### Preprocessing the genomic ground truth

The raw reads of the low-pass WGS MCF7 cell line[52], as well as the WES reads of the BCC SU006 (pre- and posttreatment)[35], called BCC06 and BCC06post in the following, and the Multiple Myeloma (sample MM199)[36], were first trimmed in order to remove low quality reads, using Trimmomatic (v0.39) (removing the first and last bases with quality less than 5, sliding window of size 4 and cumulative quality of 15, removing reads with length less than 36 bases). The trimmed reads were aligned to the human genome, version hg38, using BWA[53] (v0.7.17). The aligned reads were filtered for duplicate reads (PCR artifacts), using the MarkDuplicates software from the Picard tools[49] (v3.0.0). For all the WES tumor-control paired samples, CNVs were called using GATK4[54] (v4.4.0.0). Best Practices recommendations[55,56], tuned for WES datasets. For the low-pass WGS MCF7 sample, CNVs were identified using ichorCNA[57] (v0.2.0) and in accordance with the tool manual.

The CNVs of the COLO320 WGS dataset[58] was performed using CNVkit[59] (v0.9.10), with parameters set for WGS as suggested in the manual.

For the gastric cell lines[33] and the mouse dataset generated in this study, the scWGS ground truth copy numbers were called with the same process as in the publication[16] for the SNU601 cell line using Aneufinder[60] (v1.30.0). For the mouse data, the genome and blacklist files of mm10 were used.

The scWGS data for the iAMP21[61] dataset was initially downloaded aligned to the hg19 genome version. In order to be consistent with the scRNAseq dataset, we first converted the bam files back to unaligned fastq files, using the bamtofastq command from the bedtools[62] suite (v2.28.0), after sorting each bam file by read name, to maintain the pairs in correct order, using samtools[46] (v1.8). For each cell we used BWA[53] (v0.7.17) to align to the hg38 genome. The copy numbers were then called the same way as for the scWGS of the gastric cell lines.

For both cell lines of the DNTR-seq dataset[29], the scWGS part of the multiome was processed as described in the publication. Briefly, the sequencing data per cell was corrected for quality, using Trimmomatic (v0.39), then the reads were aligned to the hg38 genome, using BWA (v0.7.17) and finally the aligned reads were filtered for duplicates, using the MarkDuplicates software from the Picard tools (v3.0.0). After the reads were prepared, Aneufinder (v1.30.0) was used to call the groundtruth copy numbers per cell. For two ALL samples from the same dataset, the copy numbers of the groundtruth were given directly from the authors of the publication[29].

### Evaluation metrics for comparison to genomic ground truth

Different approaches were implemented to compare the continuous CNV estimates from the CNV scRNA-seq methods with the genomic ground truth datasets. Before the evaluation, the CNV scores from each scRNA-seq method are aggregated to pseudobulk scores, combining all cells annotated as tumor cells. In case the genomic ground truth was also generated from single-cell data, here the pseudobulk is calculated. Genomic CNV results were obtained in 100kB bins. The CNV scores from the scRNA-seq methods exist in different formats, either per gene or for longer segments. Each version was mapped to the 100 kB bins by calculating overlaps with the R package GenomicRanges[63] (v1.46.1). If multiple genes or segments overlapped with one 100kB bin, the average score was taken. This resulted in two vectors of continuous CNV scores per bin, one for the genomic ground truth and one for the CNV estimates.

We chose several threshold independent metrics for the benchmarking, namely Pearson correlation between scRNA-seq CNV calls and the genomic ground truth, and different versions of the AUC. For the AUC scores, an AUC (gain) was calculated for predicting gain versus the other CNV types (base or loss) and an AUC (loss) for predicting loss versus the other CNV types (gain or base) using the R package ROCR[64] (v1.0-11).

However, not all thresholds are reasonable for gain or loss prediction, respectively. Every method has a baseline value, usually 0, and only loss values lower than the baseline and gain values higher than the baseline are biologically reasonable. Therefore, we calculated additionally a partial AUC restricted for sensitivities between 0 and s_max using the R package pROC (v1.18.2)[65]. s_max is thereby defined by the baseline. We reported the partial AUC loss and gain values on top of the standard AUC values.

The interpretation of partial AUC values below 0.5 is different compared to classic AUC values. For classic AUC analyses, values close to 0 suggest swapped class labels, which we however never observed for the CNV predictions. For partial AUC values, it usually shows that the baseline level was assigned wrong, so that most thresholds were outside the biologically meaningful value range.

For the AUC calculations, a discrete ground truth is required. For the bulk data, this is given directly, for the scWGS, loss values are defined by at least 50% of the cells having a loss at this segment and gain values as at least 50% of cells having a gain there.

Furthermore, to obtain sensitivity and specificity values for gains and losses, we evaluated the optimal gain and loss thresholds, respectively. For each method, the multi-class F1 score was calculated for each possible threshold combination, again limiting the search to biologically meaningful gain and loss values. The multi-class F1 threshold was estimated based on the mean of the gain F1 score, the base F1 score and the loss F1 score using the R package crfsuite (v0.4.2)[66]. By giving each CNV the same weight, the identification of rare CNV classes also remains important.

We compared the performance of the methods to determine if they were significantly different from each other, using two-sided Wilcoxon signed-rank tests for the four main metrics (maximal F1 score, correlation, partial AUC (gain) and partial AUC (losses), and adjusted the p-values using Benjamini–Hochberg (BH). A similar

approach was chosen to test whether the methods are more closely related to each other than the ground truth. For this, we compared the correlation of the top-performing method Numbat (Expr) with the other methods to the correlation of Numbat (Expr) with the genomic ground truth, using one-sided Wilcoxon signed-rank tests (greater than) and BH-adjustment for p-values.

## Analyzing the performance for focal CNVs

We analyzed the performance to identify focal CNVs for all human droplet-based datasets, for which a scWGS ground truth dataset was available. As before, the CNV calls from the scWGS were first discretized: loss values are defined by at least 50% of the cells having a loss at this segment, and gain values are defined by at least 50% of cells having a gain there. Next, neighbouring segments with the same CNV status were combined. All CNVs (gains or losses) smaller than 3MB were defined as focal type 1. Next, the overlap between CNVs and chromosome arms was calculated. CNVs that lie completely within a chromosome arm and have additionally a size smaller than 50% or 95% of the overlapping chromosome arm, respectively, were defined as focal CNV types 2 and 3. Then, the sensitivity of detecting gains and losses for each of the focal classes was calculated with the same approach as for the general CNVs, taking the same optimal gain and loss thresholds estimated before for the general CNVs.

## Exploring performance differences across datasets

Each tested cancer dataset was described with different characteristics. Several values were taken from the scRNA-seq count matrices: the number of cells in the dataset, the mean number of UMI per cell, the total number of UMI in the dataset, the mean number of expressed (i.e., non-zero) genes per cell, the mean dropout rate per cell, and the mean coefficient of variation across genes. The coefficient of variation is calculated based on the standard deviation $\sigma$ of the log-normalized counts with the formula[67] $\sqrt{e^{\sigma^2} - 1}$.

Specifically for the evaluation of Numbat and CaSpER, the identification of SNPs in each dataset was also explored: the total number of identified SNPs by CaSpER and Numbat, respectively, as well as the number of cells with at least one SNP according to Numbat. CaSpER identifies the SNPs only per sample, not separately in each cell.

From the genomic ground truth data, we extracted several characteristics to describe the expected CNV distribution: the fraction of loss bins and gain bins from all annotated bins, the fraction of CNV bins of any type, i.e., loss or gain, and the number of breakpoints, where the CNV status changes. A higher number of breakpoints shows that there are more (and potentially shorter) CNV segments. Lastly, we included the total number of genomic bins, each of size 100 kB, which were evaluated for each dataset.

We evaluated the impact of all dataset characteristics on the performance by calculating the Pearson correlation between each feature and the maximal F1 score.

## Evaluation of diploid samples

We evaluated how well the CNV prediction methods perform on diploid datasets by testing them on CD4 + T cells from a PBMC dataset of a healthy donor[24]. First, we annotated the cell types in the dataset using classical marker genes, following the muon tutorial[68]. Then, we randomly split the CD4 + T cells and used 50% as a test dataset for the prediction in combination with four different reference datasets. As references, we tested the other 50% of the CD4 + T cells and the CD14+ Monocytes from the same dataset, as well as CD4 + T cells and CD14+ Monocytes from a second PBMC dataset[30]. As the second dataset was originally mapped with an old version of CellRanger (v1.0), which did not count intronic reads, we remapped it with a newer version of CellRanger (v7.0). We additionally tested the differences when using the newly mapped matrix and the original published matrix from GEO.

For the evaluation, we assume a completely diploid ground truth, i.e., every bin is base. This requires different evaluation metrics than for the cancer datasets. Here, we calculated the RMSE of each method from the zero baseline. As the magnitude of the scores is different, we first normalized the scores based on their standard deviation in the SNU601 dataset. This way, we can interpret the deviation dependent on a typical deviation seen in a cancer dataset.

We tested whether there were significant differences between the results based on the new and the old version of CellRanger by applying a paired $T$ test. For this, we confirmed the normality of the RMSE values using QQ-plots and Shapiro-Wilk tests.

## Performance of aneuploid datasets with different reference sets

We tested for two of the cancer datasets different reference datasets, SNU601 and MM. For each reference dataset, we repeated the whole evaluation workflow, i.e. running every method and comparing the results to the genome ground truth. For the MM dataset, we tested different immune blood cells as reference, taking the diploid dataset we evaluated before[24]. We tested three different subsets, T cells, B cells and Monocytes. Additionally, we tested two extreme references, the SNU601 dataset[33] and the gastric reference of the SNU601[32].

For SNU601, we tested a second gastric dataset[34], which we split into three subsets dependent on the published cell type annotation. The first subset contains epithelial and endothelial cells, the second subset fibroblasts and smooth muscle cells and the third dataset all types of immune cells. Additionally, we tested the SNU601, taking the cancer cells from the MM dataset as a reference, an extreme case where the reference itself contains CNVs.

For both datasets (MM and SNU601), we tested whether there are significant differences between the datasets. We performed a Wilcoxon signed rank test between the originally chosen reference dataset and all other tested reference datasets, one-sided (whether the original chosen is better), corrected for multiple testing with FDR.

## Evaluation of automatic cancer cell identification

There are four scRNA-seq CNV calling methods which can be run without providing an explicit cell type annotation: CONICSmat, CopyKat, Numbat and SCEVAN. In the previous analyses, they were always run with a defined reference annotation to increase the comparability with the other methods. Here, each was tested a second time without explicit reference cells for the four primary tumor datasets (MM, iAMP21, BCC06 and BCC06post). These four datasets comprise a mixture of tumor and euploid cells. The performance was evaluated with the same metrics as in the previous analyses, and the differences with and without the explicit reference cells were compared.

Three of the methods (CopyKat, Numbat and SCEVAN) additionally annotate the cells as cancer cells vs euploid cells when running without an explicit annotation. We compared these annotations to our own manual cell type annotations for the MM dataset and the iAMP21 dataset and to the published cell type annotations for the two BCC datasets.

## Evaluation of the identified subclonal structure

A known ground truth was required to evaluate how well each method can distinguish different CNV clones. For this, the pre-treatment data of four donors of the BCC dataset were chosen, selecting the donors with at least 20 cancer cells. To maintain consistent naming throughout the manuscript, the samples, originally called su00x in their publication, are called BCC0x here. First, we ran each donor separately with copyKat and clustered the per-cell CNV predictions afterward to verify that the donors have indeed distinguishable CNV profiles, i.e., can be seen as separate clones.

Then, a combined count matrix and BAM file were generated to mix the four samples, excluding all donor information in the analysis. Afterward, the standard CNV calling pipeline was run. Numbat,

InferCNV and SCEVAN directly provide a classification of cells into distinct groups, i.e., subclones, while CopyKat, CONICSmat and CaSpER return only a hierarchical clustering. For the last three methods, we cut the tree at a height of 4 to obtain distinct groups for the evaluation.

We used two different metrics to compare the clusters to the ground truth of patient annotations, the adjusted Rand Index (ARI) using the R package mclust (v6.1.1)[69] and the Homogeneity score using the package clevr (v0.1.2)[70]. The ARI evaluates the similarity between the two clustering results. However, some of the methods split the cells of one patient into multiple clusters, so potentially multiple subclones per patient, which we can not verify with this dataset. This problem is overcome with the Homogeneity score, which evaluates whether all cells from one cluster belong to the same group, here to the same patient. Splitting one patient into several subclones is thereby not punished.

Of note, analyzing different donors with different SNP profiles together is not the recommended approach for the AF-based methods CaSpER and Numbat. Differences in heterozygous sites between donors are an additional unwanted source of variation not related to cancer mutations. Nevertheless, Numbat was capable of performing very well in the analyses above, with the same accuracy as purely expression-based methods.

### Statistics & reproducibility
No statistical method was used to predefine the sample size. We chose the datasets to reflect different cancer types, scRNA-seq technologies and organisms, as well as a diploid dataset to get a broad representation of use cases for scRNA-seq CNV callers. From the SNU601 scWGS dataset, a set of 134 cells was excluded from the corresponding comparison because of their very high ploidy, which did not align with the majority of the cells. Since the cell lines were measured in different laboratories, clonal variation among them can be expected. No randomization and blinding took place in the study; we used all available datasets.

We applied Wilcoxon tests to compare the performance metrics, further details about the different statistical analyses can be found in the respective method sections. All CNV callers were tested on 20 different cancer datasets to ensure reproducibility of the performance metrics. We provide a snakemake pipeline to ensure reproducibility for other users.

### Reporting summary
Further information on research design is available in the Nature Portfolio Reporting Summary linked to this article.

### Data availability
The mouse data generated in this study have been deposited in the Biostudies database under accession code S-BSST1928. The previously published datasets used in this study are available here: For the nine gastric cell lines[33], the scWGS data were downloaded from the SRA repository, with accession number PRJNA498809. The corresponding scRNA-seq data were downloaded from the SRA repository PRJNA598203, and the control samples for the scRNA-seq[32] data from the GEO repository with accession number GSE150290. A second gastric dataset[34] was tested as an alternative reference for the SNU601, downloaded from GEO (accession number GSE159929, the stomach dataset GSM4850590). For the breast cancer cell line MCF7[52], ultra-low pass whole-genome sequencing data were downloaded from the ENA, project PRJNA398960 (Biosample SAMN07519582), while the scRNA-seq dataset was downloaded from GEO (accession number GSM3142233). As a reference for the MCF7 cell line, the mammary gland dataset from the Tabula Sapiens cell atlas[71] was used [https://figshare.com/articles/dataset/Tabula_Sapiens_release_1_0/14267219]. The MM whole exome sequencing dataset[36] was downloaded from the

GEO (accession number GSM4200481 for the control sample [https://www.ncbi.nlm.nih.gov/sra/?term=GSM4200481] and GSM4200480 for the tumor sample), while the corresponding scRNA-seq dataset was downloaded from GEO (accession number GSM4200471). The COLO320HSR whole-genome sequencing dataset[58] was downloaded from the SRA accession PRJNA506071 (sample SRS4831935 [https://www.ncbi.nlm.nih.gov/sra/SRX5930165[accn]]), the multiome data[72] of the same cell line from the SRA accession PRJNA672109 (sample SRS7587918), and the control scRNA-seq samples[73] were acquired from the gut cell atlas [https://www.gutcellatlas.org/]. The BCC[35] scRNA-seq samples were downloaded from the GEO repository with the number GSE123814 (GSM3511758 for su006 and GSM3511761 for su006 post, GSM3511753 for su005, GSM3511763 for su007 and GSM3511767 for su008). For the BCC sample su006 WES data were downloaded from the SRA accession PRJNA533341 (sample SRS4645189 [https://www.ncbi.nlm.nih.gov/sra/SRX5705755[accn]]). The two DNTR-seq cell lines[29] (HCT116 and A375) were downloaded from the GEO repository, accession number GSE144296. The respective reference cells for the HCT116 cell line[74] were taken from the GEO repository with accession number GSE95435, and the reference cells for the A375 cell line[75] from GEO with accession number GSE151091. The two primary samples of the DNTR-seq data (ALL1 and ALL2) were obtained directly from the authors[29]. Access to these datasets is restricted, due to being primary patient data, but can be requested from the corresponding author, Dr. med. Martin Engee. The primary sample of ALL with intrachromosomal amplification in chromosome 21 (iAMP21)[61] was downloaded from the European Genome-Phenome Archive (EGA). The accession number of the scWGS sample is EGAD00001010288, and the corresponding scRNA sample is EGAD00001009504. Access to these datasets is restricted, as they are primary patient data. Access can be requested directly through the above links. The two PBMC datasets were downloaded from 10x Genomics [https://www.10xgenomics.com/datasets/pbmc-from-a-healthy-donor-no-cell-sorting−10-k-1-standard-2-0-0] and GEO accession GSE96583[30] (sample GSM2560248, batch A). Source data are provided with this paper.

### Code availability
All code written for this study, especially the benchmarking pipeline with snakemake, can be found on GitHub: https://github.com/colomemaria/benchmark_scrnaseq_cnv_callers[21].

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

## Acknowledgements

We thank Dr. Gregor Miller and Ronan Le Gleut from the Core Facility Statistical Consulting at Helmholtz Munich for statistical advice. We thank the BMC Bioinformatics Core Facility for providing access to their HPC cluster. We thank Martin Enge and Vasilios Zachariadis for providing the DNTR-seq data from the ALL patients. This work is funded by the Deutsche Forschungsgemeinschaft (DFG, German Research Foundation)—Projektnummer 553739126 for M.C.T. This work was supported by HelmholtzAI for AS and was funded by the Deutsche Forschungsgemeinschaft (DFG, German Research Foundation)—Project-ID 213249687—SFB 1064 for DH. FF received funding from the Dutch Cancer Society grant 2018-RUG-11457.

## Author contributions

M.C.T. designed the study. K.T.S., A.S., and M.L.R. prepared the data. K.T.S. and D.H. implemented the benchmarking pipeline. A.E.T. and F.F. generated the mouse dataset. K.T.S. and M.C.T. wrote the manuscript with support from A.S. and M.L.R. All authors reviewed the final manuscript.

## Funding

## Competing interests

F.F. is the chief scientific officer of iPsomics, a company that commercializes single-cell genomics. The remaining authors declare no competing interests.
