## [Transparent Peer Review file · Nature Communications]

Benchmarking scRNA-seq copy number variation callers

Corresponding Author: Professor Maria Colomé-Tatché

Version 0:

Reviewer comments:

Reviewer #1

(Remarks to the Author)

Single cell whole-genome sequencing is considered the gold-standard for the quantification of CNVs in single cells. However, the majority of existing single cell datasets interrogate gene expression, using scRNA-seq. For this reason, Schmid et al presented the first benchmark of 6 software programs calling somatic CNVs from single-cell RNAseq. They used 15 scRNA-seq datasets and evaluated six popular computational methods to predict CNVs versus euploid cells, in fully diploid samples and heterogeneous tumor samples.

They state that while these algorithms relied on different approaches, their predictive power is similar and is depended on dataset quality. Finally, they did an thorough overview of the different functionalities of the software programs, showing that further choices of such algorithm should rely more on the options offered by the algorithms than on their predictive performances which are similar.

My major concern is the lack of statistical tests. In particular, authors do not provide any statistical comparison between the performances of the algorithms reviewed in the paper. As a result, most of the findings fail to convince the reader, especially when they state that some methods are better than others.

There are some confusing/false interpretation of the AUC scores for Numbat. An AUC close to 0 is an excellent predictor (it's a perfect exclusion). AUC should be expressed between 50 and 100.

Page 10: the effect of the references between euploid and aneuploid cells can't be compared because two different metrics are being used.

Page 3, There is no description of the ground truth (Nbr CNVs, gain loss etc.)

Page 4, What criterion was used to annotate cancer and healthy cells ? What euploid dataset was used ? Is the CNV distribution derived from the scWGS ?

Page 6, Can the authors detail the rationale for selecting such features in figure 2.B ? The authors need to provide some statistics to test if the CNV callers predictions tend to agree more between methods than when compared to the genetic ground truth.

What is the expression threshold mentioned in supp table 2?

In Supp Fig 9 and 10, the differences between the cell and the subclones seem to be significant, but it is unclear know if we can say small.

Page 8, what is acronym "AF" ? What is the range of the proportion of non-diploid genomes across method and datasets, and are these biased toward gain or loss?

In Supp Figure 12, the performance doesn't seem to be significantly decreased. Can the authors test this statistically ?

In Supp 13 what was the reference used to compute the deviation ?

Page 11, named the paper with the ref 21

Is the drop really significant when using SNU601, and different from other references?

Page 14, authors mention that some methods used multithreading but mention only Numbat.

Page 15, It is unclear what results suggest that that loss were more difficult to identify than gains?

Page 17: " Bam files and count matrices were generated from the fastq files by running CellRanger (version 7.0.0), except for the BCC dataset, where the annotated count matrix was taken directly from GEO." For the BCC dataset, you mention another Cell ranger version without giving the version.

(Remarks on code availability)

NA.

Reviewer #2

(Remarks to the Author)

This study examines a topical and interesting area in bioinformatics - inference of copy number from single cell RNA seq data. The authors have done an extensive job in evaluating multiple callers and presented comparative metrics and made some recommendations. I think this potentially is a valuable study with revision - it is an area that this reviewer is interested in and I was not aware of the pros and cons of many of these callers. I have a couple of major suggestions for improvement.

The datasets used could be improved. They are mostly cell lines, unpaired, and even the cancer dataset apparently is compromised by the lack of paired, single cell and bulk reference WGS data. There are more comprehensive real world datasets that are publicly available - one is PMID 37216686 (extensive paired WGS with complex copy number alterations, single cell WGS and single cell RNA-seq). Others exist (e.g. medulloblastoma data from the Northcott group at St Jude). I would hesitate to move ahead with the recommendations presented here without a detailed benchmarking of optimal datasets. I would also like to see analysis of ability to resolve focal copy number alterations. That is not addressed here. Many of the visualizations refer to statistical accuracy, or genome-wide profiles, and could be improved.

Some of the callers (e.g. Numbat) appear excessively clean and may undersegment data (compared to WGS for example). This is related to the focal CNA point above.

(Remarks on code availability)

I do not have time.

Version 1:

Reviewer comments:

Reviewer #1

(Remarks to the Author)

Authors have carefully responded to most of our comments.

We have reviewed the Github repository (https://github.com/colomemaria/benchmark_scrnaseq_cnv_callers).

The README is clear and provides detailed information. However, we have not attempted to implement these pipelines.

We urge the authors to remove the truncated AUC method from the manuscript. This method is not intuitive and may not be well-received by the community. AUC has been around forever, and it is never truncated.

If some values are inappropriate to predict a deletion or a duplication, they should not be integrated into the predictive model. If aberrant values can be clearly defined beforehand, they can also be removed from the predictive model.

(Remarks on code availability)

Reviewer #2

(Remarks to the Author)

Overall, the authors have been highly responsive to the suggestions of this reviewer, including analysis and benchmarking of callers across multiple additional datasets. This was the main area for improvement.

A couple of extra points if the manuscript proceeds:

Be careful about designating "copy number" subclones based on subchromosomal changes on chromosome 6, particularly 6p (the part of the rebuttal and manuscript that refers to SFig 20). This can be a pseudo CN change due to coordinate change in expression of histone genes due to a subpopulation of cells being in cell cycle - not a real copy number based subclone.

My suggestion about improving visualization referred to the fact that the number of figures in the main MS is not that great, and many refer to benchmarking/statistical analyses rather than depictions of actual regions/samples/data to get a good feeling for real world performance of the callers.

(Remarks on code availability)

Reviewer #3

(Remarks to the Author)

(Remarks on code availability)

Version 2:

Reviewer comments:

Reviewer #1

(Remarks to the Author)

Authors have responded to our comments.

They improved the description and explanation of the partial AUC method.

However, this approach will make comparing AUC across studies more difficult (Chaibub Neto et al. 2024. "A Novel Estimator for the Two-Way Partial AUC." BMC Medical Informatics and Decision Making 24 (1): 57. <https://doi.org/10.1186/s12911-023-02382-2>).

(Remarks on code availability)

Reviewer #2

(Remarks to the Author)

The authors have addressed my comments

(Remarks on code availability)

Reviewer #3

(Remarks to the Author)

(Remarks on code availability)

Point-by-point response to the reviewers

We would like to thank both reviewers for their valuable comments and for having taken the time to review our work. We found the reviewers' comments helpful and believe that the changes we have made based on their feedback have strengthened our study.

Following the suggestions of reviewer 1, we have included several statistical analyses to support our interpretations. Additionally, we followed the comments of reviewer 2 to include more datasets and a specific analysis of focal CNVs. Our general results from the first submission still hold, but we could add several additional interesting insights, such as performance differences between droplet-based and plate-based datasets and performance differences depending on the size of the CNVs.

During the evaluation of the new datasets, we corrected some minor inconsistencies in our old evaluation pipeline: for inferCNV, the runtime was not tracked correctly, for CONICSmatrix, we updated the gene annotation version, and for SCEVAN, we filtered out cells that were annotated as cancer by SCEVAN, but normal in our initial annotation, so that all methods are compared on the same set of cells. We also updated the search area of the maximal F1 scores to include now also the baseline value as an additional threshold, which is relevant in very few extreme cases. None of the changes resulted in major differences to the previous version.

In the following, we provide a point by point responses for all comments.

Reviewer comments are marked in green and italic font.

Responses are typeset in regular font.

Edits in the main text are marked in brown.

Reviewer #1 (Remarks to the Author):

Single cell whole-genome sequencing is considered the gold-standard for the quantification of CNVs in single cells. However, the majority of existing single cell datasets interrogate gene expression, using scRNA-seq. For this reason, Schmid et al presented the first benchmark of 6 software programs calling somatic CNVs from single-cell RNAseq. They used 15 scRNA-seq datasets and evaluated six popular computational methods to predict CNVs versus euploid cells, in fully diploid samples and heterogeneous tumor samples.

They state that while these algorithms relied on different approaches, their predictive power is similar and is depended on dataset quality. Finally, they did an thorough overview of the different functionalities of the software programs, showing that further choices of such algorithm should rely more on the options offered by the algorithms than on their predictive performances which are similar.

My major concern is the lack of statistical tests. In particular, authors do not provide any statistical comparison between the performances of the algorithms reviewed in the paper. As a result, most of the findings fail to convince the reader, especially when they state that some methods are better than others.

We thank the reviewer for the suggestion to add more statistical analyses for the interpretation, which have made our results more robust. We compared the performance between methods using a Wilcoxon rank sum test (two sided), pairwise for each method pair across the four main metrics and adjusted for multiple testing using Benjamini Hochberg. The Figure below shows the resulting p-values, significant p-values are highlighted in green. We updated the respective results section to incorporate these results:

“We evaluated all CNV callers on 44 15 different human cancer datasets measured with droplet-based scRNA-seq technologies using various metrics. On average, Numbat (Expr), copyKat and InferCNV (Expr) had the highest maximal F1 scores (between 0.59 to 0.57), and also scored high for all other metrics (**Figure 2A, Supplementary Figure 4**). However, the performance differences between the callers were in most cases non-significant, when considering the maximal F1 scores (**Supplementary Figure 5**). More significant differences were visible for the correlation and truncated AUC scores. Furthermore, ~~However, performance differences between most of the callers on the same dataset remained small, and~~ all metric scores for all methods showed a large standard deviation across datasets (**Figure 2A, Supplementary Figure 4**). Due to these aspects ~~this large spread~~, no method can be seen as clearly superior to the others”

New Supplementary Figure 5. FDR-corrected p-values from Wilcoxon signed rank tests (two-sided) to compare the method’s performance across droplet-based human datasets, pairwise for each combination of methods.

And in the methods section, we added the following information:

“We compared the performance of the methods, to determine if they were significantly different from each other, using two-sided Wilcoxon sign-rank tests for the four main metrics (maximal F1 score, correlation, truncated AUC (gain) and truncated AUC (losses), adjusted the p-values using Benjamini-Hochberg (BH).”

There are some confusing/false interpretation of the AUC scores for Numbat. An AUC close to 0 is an excellent predictor (it's a perfect exclusion). AUC should be expressed between 50 and 100.

We use in most cases not the classical AUC definition, but what we call “truncated AUC”, for example in Figure 2A which the reviewer is likely talking about here. We acknowledge that this metric, which was developed by us specifically for this benchmarking, requires more introduction and especially an explanation why the value range is different from a standard AUC value.

The classical AUC is designed for a binary classification problem, while the CNV classification contains three classes. So even if we have a continuous score for the CNVs, every method has a baseline value (for the “diploid” state) and not every value is reasonable to be tested as a gain / loss threshold. For example for copyKat, only values larger than 0 should be tested when evaluating the gains and only values smaller than 0 should be tested when evaluating the losses. In the classical AUC definition, this is not taken into account, and every value is tested. To exemplify this, we show below (**new Supplementary Figure 2**) the ROC for gains in the dataset MCF7, using copyKat, and testing all threshold values (standard AUC). Here, we can see an AUC value of 0.87. However, the upper part of the AUC (grey area) is based on negative thresholds, which define the loss class in copyKat and not the gain class. Therefore, we calculate only the AUC of the yellow area until the threshold of 0, which gives an truncated AUC of 0.60.

New Supplementary Figure 2. Illustration of the truncated AUC scores, exemplarily shown here for the ROC curve for CopyKat, for identified gains applied to the MCF7 dataset. The area under the curve (AUC) is only calculated for thresholds which are inside the allowed value range for the CNV type and method. Here, for gains the only allowed thresholds are above 0 (for losses the allowed thresholds would be below 0). The vertical line visualizes the sensitivity when using 0 as threshold.

Therefore, the truncated AUC shows whether the baseline is estimated correctly, something that the methods have problems with in some datasets, for example for the MCF7 dataset

(see **Supplementary Figure 9J**) - in these cases there are larger deviations between the AUCs and the truncated AUCs (**Supplementary Figure 4**).

The truncation at the baseline threshold shifts also the interpretation of the AUCs, now values between 0 and 1 are possible. A value of 0, as happens in some Numbat scenarios, means now that the baseline prediction was completely off. For example for KATOIII, Numbat estimates for all loss regions (according to the WGS ground truth) a CNV value larger than 0 (corresponding to base or gain status) and no ROC curve can be calculated (see also **Supplementary Figure 5 D**).

The reviewer is right that in the classical AUC interpretation, a value close to 0 would indicate that the classes are swapped, i.e. gain regions predicted as losses and vice versa. The same interpretation can not be made from the truncated AUCs anymore, but we also report the classic AUC values in **Supplementary Figure 4**. As seen there, the AUCs are never close to 0, but in some cases lower than 0.5. We would nevertheless argue that this should be reported like this (and not only values above 0.5), because a wrong assignment of class labels is nevertheless a mistake of the tools, which should be indicated.

We updated the manuscript with a new Supplementary Figure 2 (shown above) to explain the truncated AUCs better. We added in the results section:

“Not the complete range of thresholds is biologically **reasonable—meaningful** to classify regions as gains or losses, respectively, as every method defines a baseline score. For this reason, we chose to implement a truncated version of the AUC scores where only thresholds up to the baseline score were evaluated for losses and only scores higher than the baseline score were evaluated for gains (see **Methods, Supplementary Figure 2**). **Of note, truncated AUC values below 0.5 indicate that most thresholds are outside the biologically meaningful value range (see Methods).**”

And in the methods section:

“**The interpretation of truncated AUC values below 0.5 is different compared to classic AUC values. For classic AUC analyses, values close to 0 suggest swapped class labels, which we however never observed for the CNV predictions. For truncated AUC values, it usually shows that the baseline level was assigned wrong, so that most thresholds were outside the biologically meaningful value range.**”

Page 10: the effect of the references between euploid and aneuploid cells can't be compared because two different metrics are being used.

We thank the reviewer for pointing out this issue. Unfortunately, it is not possible to use the same set of metrics for both data types, as the calculation of correlation, AUC etc does not work if we assume a completely diploid baseline. We removed the comparison between euploid and aneuploid cells completely from the text and interpret both separately now.

Page 3, There is no description of the ground truth (Nbr CNVs, gain loss etc.)

We thank the reviewer for pointing this out. We added now a short description and a separate Supplementary Figure 1 for this:

Supplementary Figure 1. CNV distribution across datasets. The CNV type is assigned based on the genomic ground truth ((sc)WGS or WES). The distribution is shown both A. for the CNVs in the whole genome, i.e. the whole area for which we have genomic information available for this dataset, and B. for the CNVs in the area of the genome which is covered by the scRNA-seq CNV callers.

And we added the following text to the first result section of the manuscript:

“Different metrics were included for benchmarking, most of them based on the comparison with a ground truth for the CNVs. We obtained this ground truth from either (sc)WGS or WES data (**Supplementary Table 1**). The different datasets showed large variation in CNV distribution, with CNVs covering between 7% to 93% of the total genome and more gained regions than lost regions in the majority of the datasets (**Supplementary Figure 1**).”

Page 4, What criterion was used to annotate cancer and healthy cells? What euploid dataset was used? Is the CNV distribution derived from the scWGS?

Question part 1 - criterion used to annotate cancer and healthy cells:

We apologize that the cell type annotation description was not clear enough. We have now improved the description on the main text:

“To ensure reproducibility between methods, cells were annotated manually into tumor and healthy cells per sample, and the same healthy cells were used as reference for all methods unless specified otherwise. We applied the published cell type annotation for the BCC and ALL datasets and performed manual annotation based on Louvain clustering and known marker genes for the MM, the iAMP21 and the mouse datasets (see **Methods**).”

This is the respective methods section:

“The cell type annotation is only relevant for the primary cancer samples, as the cell lines are assumed to be a homogenous set of cells and all labeled as the cell line. For the BCC dataset, the published cell type annotation was used. For the MM dataset, a manual annotation was performed, using Louvain clustering and known marker genes for MM (CD38, IGHM, MZB1), *after performing some standard filterings steps (removing cells with low counts or high mitochondrial fraction)*. The same way, a manual cell type annotation of the iAMP21 dataset was done with the following key annotation marker genes: MS4A1 for B cells, HBD for erythroid cells, CD3D for T and NK cells and TNFRSF17 for the cancer cells. The aneuploid T cell lymphoma from mouse (T989) was annotated the same way. It contained mostly T cells (identified using Cd3e as marker) and a small fraction of B cells (using Cd74 & Itgax as markers). We removed the B cells, so that the dataset contained only aneuploid T cells. As a reference for this dataset, the count matrix of the second euploid T cell lymphoma (eT) was used, again after basic quality control.”

And further down in the same methods section:

“The two primary DNTR-seq ALL samples were obtained already processed, including a cell type annotation.”

Question part 2 - What euploid dataset was used ?

Thank you for pointing this out, we added this information for clarification:

“For this reason, we included an euploid dataset, *comprising PBMCs from a healthy donor²¹*, in our performance test and calculated the mean square error deviation for every method compared to a diploid reference genome.”

Question part 3 - Is the CNV distribution derived from the scWGS ?

Yes. To make this clearer, we have updated the figure legend (now Supplementary figure 1) accordingly and would like to thank the reviewer for pointing out that this was not documented well enough in the previous Supplementary Figure 3:

“Supplementary Figure 1. CNV distribution across datasets. The CNV type is assigned based on the genomic ground truth ((sc)WGS or WES). The distribution is shown both A. for the CNVs in the whole genome, i.e. the whole area for which we have genomic information available for this dataset, and B. for the CNVs in the area of the genome which is covered by the scRNA-seq CNV callers.”

Page 6, Can the authors detail the rationale for selecting such features in figure 2.B ? The authors need to provide some statistics to test if the CNV callers predictions tend to agree more between methods than when compared to the genetic ground truth.

Question part 1 - rationale for selecting features in Figure 2B

We thank the reviewer for this comment and clarified our choice for features in Figure 2B. The full set of tested features is shown in Supplementary Figure 8. We split this

Supplementary Figure now into part A), general dataset characteristics and B) dataset characteristics associated with SNP calling, which are only relevant for CaSpER and Numbat. In the main Figure 2B, we selected the top 7 general dataset characteristics features, which have the largest effect on the performance. This was estimated by the highest mean absolute correlation values across methods. We added this to the figure legend to explain our choice:

“Impact of dataset characteristics on the performance (maximal F1 scores). The features here show the total number of UMI counts, the number of cells **and the number of expressed genes** in the cancer dataset, the mean dropout rate per cell, the mean coefficient of variation across genes, the fraction of gained regions from all ground truth annotated regions **and the fraction of CNV regions (gained and lost together)**. These are the top 7 general dataset characteristics with the highest mean absolute correlation to the performance, all features are shown in the supplement (**Supplementary Figure 8**).”

We added the fraction of total CNV regions to Figure 2B, to make the feature selection more consistent. However, we want to note that this feature is strongly correlated with the feature “fraction of gained regions” (cor = 93%). We added a respective remark in the text to clarify this:

“The strongest negative correlation however was observed with the fraction of genomic aberrations: the larger the fraction of the genome in the gain state, the more difficulties all methods had to infer the correct CNV profile. This is probably due to the fact that all methods seemed to have a problem identifying the baseline ploidy in these extreme cases (**Supplementary Figure 1+9**). The same trend was also visible for the fraction of CNV regions in general, which is however strongly correlated with the fraction of gained regions for the tested datasets.”

We also added the number of expressed genes to Figure 2B, because with the addition of more datasets (see answer to reviewer 2 for details), the feature importance shifted slightly and both “the number of expressed genes” and the “coefficient of variation” became very similarly correlated to the performance (F1 score).

Question part 2 - statistical test for proof that methods are more similar to each other compared to the ground truth

To show that the methods are more similar to each other than to the ground truth, we perform a one-sided Wilcoxon rank signed test, to test whether the correlation between the best performing method (“Numbat (Expr)”) and any other method is greater than the correlation between “Numbat (Expr)”) and the genomic ground truth. The p-values were adjusted for multiple testing using Benjamini-Hochberg.

The adjusted p-values were highly significant (p-value < 0.005) for three other methods, “InferCNV (CNV)”, “InferCNV (Expr)” and “copyKat” and significant (p-value < 0.05) for two more, “SCEVAN (CNV)” and “SCEVAN (Expr)”.

This shows that part of the methods are indeed more similar to each other than compared to the genomic ground truth.

We thank the reviewer for this suggested analysis, which makes our manuscript more clear. We added a respective sentence:

“For some of the CNV callers, the predictions tended to agree more between methods than when compared to the genetic ground truth (**Figure 2C+D, Supplementary Figure 9+10**). This was in particular the case for InferCNV, copyKat, SCEVAN and Numbat (Expr) (i.e. correlation of Numbat (Expr) with InferCNV, copyKat and SCEVAN was significantly higher than with the ground truth (Wilcoxon signed rank test, FDR < 0.05)).”

We added the description of the analysis into the methods section:

“... A similar approach was chosen to test whether the methods are more closely related to each other than the ground truth. For this, we compared the correlation of the top-performing method Numbat (Expr) with the other methods to the correlation of Numbat (Expr) with the genomic ground truth, using one-sided Wilcoxon sign-rank tests (“greater than”) and BH-adjustment for p-values.”

What is the expression threshold mentioned in supp table 2?

We apologize that this part was not clear enough. This is specifically the expression threshold as defined in one of the CNV methods, called CaSpER. We tested here two different thresholds, as the recommended threshold for 10X data from the respective tutorial seemed very lenient to us compared to the other methods (see manuscript). We extended the legend of the Supplementary Table 2 to describe this in more detail:

“**Supplementary Table 2.** Performance differences for CaSpER depending on the chosen expression threshold for the SNU601 cell line. The expression filtering is directly defined within CaSpER (function `CreateCasperObjec()`, parameter `expr.threshold`). All genes whose mean expression is smaller than the threshold are removed from the analysis. Recommended cutoff based on the 10X tutorial of CaSpER is 0.1, default parameter for bulk and plate-based single cell data is 4.5.”

The analysis becomes more clear in the results section citing this Supplementary Table:

“To show how the expression filtering influenced performance, we exemplarily ran CaSpER with a more strict cutoff on the SNU601 cell line (keeping 8,847 genes instead of 13,196 genes) and saw a clear improvement in the CNV calling results (**Supplementary Table 2**).”

In Supp Fig 9 and 10, the differences between the cell and the subclones seem to be significant, but it is unclear how we can say small.

We thank the reviewer for this comment, we extended our analysis of the per cell results in Supplementary Figure 14 to make the interpretation more precise.

We performed a one-sample Wilcoxon test for each method, comparing the pseudobulk correlation value with the per cell correlation values (one-sided to test that the pseudobulk correlation is greater, adjusted with Benjamini-Hochberg), all results are highly significant.

As shown in the plot below, the results based on subclones are closer to the pseudobulk correlation and show smaller standard deviations. This is expected, as a subclonal grouping is conceptually similar to pseudobulk estimates, but on finer granularity.

New Supplementary Figure 14. Per cell prediction performance for all methods compared to the scWGS ground-truth for the SNU601 cell line. ... **(B)** Deviation of per cell correlation values from the pseudobulk correlation value, color-coded by whether the method reports per cell results or subclonal results.

We added the Figure above as Supplementary Figure 14B and changed the respective text in the manuscript:

“Additionally to the pseudobulk evaluations, we also checked the per cell estimates. For the SNU601 cell line, the pseudobulk CNVs were closer to the ground truth compared to the per cell results for all methods, probably due to the reduction of noise (**Supplementary Figure 13+14**). We saw a performance difference between the methods which output subclone or per cell CNVs, where the first showed smaller deviation from the pseudobulk profiles. This is expected, as the subclonal aggregation is conceptually similar to the pseudobulk aggregation, but on a smaller scale. Still, independent of the approach, the mean per cell correlation was over 0.45 for all methods except CaSpER and CONICSmat. So the per cell CNV profiles of the methods are reasonable to use in downstream analysis, although an aggregation to subclonal or dataset level provides more reliable results.”

Page 8, what is acronym “AF” ? What is the range of the proportion of non-diploid genomes across method and datasets, and are these biased toward gain or loss?

Question part 1 - acronym “AF”

Thanks for pointing us to the un-introduced abbreviation AF, it stands for “allele frequency”. We know properly introduce the abbreviation when we use the term the first time in the results section:

“ and a second class that combines the expression values with minor allele frequency (AF) information, consisting of CaSpER¹¹ and Numbat¹⁴.”

Question part 2 - proportion of CNVs in non-diploid genomes

The proportion of the non-diploid genome of the aneuploid datasets (from manuscript section 2: “Benchmarking scRNA-seq CNV prediction compared to genomic ground truth”) is visualized in **new Supplementary Figure 1**. There is a large range between datasets, CNVs cover between 7% to 93% of the genome.

We account for the unbalanced class definition by choosing different metrics focusing on different aspects as described in this manuscript section 2:

“The maximal F1 score puts equal weight on predicting all three CNV classes (gain, base and loss), independent of their occurrence in the dataset. This is important as most of the datasets have more gains than losses, and some of the cell lines (HGC27, KATOIII and SNU16) have very extreme profiles with more than 75% of gained regions (**Supplementary Figure 1**). A method/dataset combination with high maximal F1 score indicates that the method was able to predict all three classes accurately.”

Whether methods are better at predicting gains or losses is further discussed at another question of the reviewer regarding page 15 (see below at page 13).

In Supp Figure 12, the performance doesn't seem to be significantly decreased. Can the authors test this statistically ?

We explored whether the use of the CellRanger version had a significant effect of the performance in general using a Paired T-Test (one-sided that the CellRanger 1 RMSE values were greater than the CellRanger7 RMSE values), separately on the CD4+ T cells (Supplementary Figure 22A) and the CD14+ Monocytes (Supplementary Figure 22B). For this, we confirmed the normality of the RMSE values using QQ-plots and Shapiro-Wilk tests. Both paired T-tests were significant with p-value=0.0038 for the CD4+ T cells and p-value=0.0008 for the CD14+ Monocytes. This confirms our statement.

We updated the manuscript text to include the new results:

“To show that, for the second reference PBMC dataset we repeated the analysis with the same data but mapped with an older version of CellRanger. This reduced the performance even more (paired T-test p-value=0.0038 for the CD4+ T cells and p-value=0.0008 for the CD14+ Monocyte; see **Methods**) (**Supplementary Figure 22**).”

And in the methods section:

“We tested whether there were significant differences between the results based on the new and the old version of CellRanger by applying a paired T-test. For this, we confirmed the normality of the RMSE values using QQ-plots and Shapiro-Wilk tests.”

In Supp 13 what was the reference used to compute the deviation ?

Thanks for pointing this out. The deviation refers to the deviation in performance between the analysis with the reference dataset initially chosen by us in the benchmarking (shown in the first column of the subplots) and the analysis with the other reference dataset in the respective column. So a positive deviation (red background color) shows that this reference dataset is performing better than the initial reference and vice versa a negative deviation (blue background color) shows that this reference dataset is performing worse. We selected

this initial reference dataset, as it was biologically the closest cell type for the normalization, therefore we see many negative deviations as expected.

We have updated the legend in the figure (now Supplementary Figure 23) accordingly:

“Supplementary Figure 23. ... The color gradient in the plot shows the deviation of each metric from the first column of the respective panel, in which the reference dataset initially chosen by us in the benchmarking is displayed. We selected this initial reference dataset, as it was biologically the closest cell type for the normalization.”

Page 11, named the paper with the ref 21

We improved the readability of this sentence:

“Here, additionally, we also tested different PBMC cell types as reference (T cells, B cells and monocytes from **10X Genomics**²¹).”

Is the drop really significant when using SNU601, and different from other references?

We thank the reviewer for the question and clarified our analysis for the best reference dataset. There are two aspects that we can look at, whether one method is more stable across different reference datasets and which is in general the best suitable reference dataset for all methods. We reordered the points in Main Figure 3B+C to highlight better the differences across reference datasets.

First, we can check how stable are the results for each method across reference datasets, i.e. which method is the best. For this, we calculate the standard deviation of the maximal F1 score per method across the datasets (see also Supplementary Figure 23). We see that these are relatively similar and relatively small (between 0.03 and 0.09) for all methods in both scenarios for the MM and the SNU601 dataset, except for SCEVAN (CNV) and SCEVAN (Expr) in the SNU601 scenario, which have some extreme outliers and a sd of 0.17 and 0.19, respectively.

Second, we can check which is the best reference dataset in general for all methods (see new Main Figure 3B+C, also copied below). For this, we performed Wilcoxon signed rank tests for the maximal F1 scores of the reference dataset initially chosen vs the other reference datasets (again FDR corrected).

Figure 3. B+C. Performance of the methods with different reference datasets for the MM dataset (B) and the SNU601 dataset (C). For the MM dataset, we tested a second healthy PBMC dataset, which we split into B cells, T cells and Monocytes (mono). We additionally tested a healthy gastric dataset and a gastric cancer dataset (SNU601). For the SNU601, we tested a second healthy gastric reference, which we split into three groups, epithelial and endothelial cells (epiendo), fibroblasts and smooth muscle cells (fibsom) and immune cells. We also tested the MM dataset as reference.

For the MM dataset, p-values are all significant (FDR < 0.05):

ref	pval	fd
MM_tcell	0.0019531	0.0024414
MM_bcell	0.0019531	0.0024414
MM_mono	0.0039063	0.0039063
MM_gastric	0.0019531	0.0024414
MM_SNU601	0.0019531	0.0024414

In contrast, the comparisons are all not-significant for the SNU601 dataset except when using the MM dataset as reference (which contains CNVs):

ref	pval	fd
SNU601_epiendo	0.3671875	0.3671875
SNU601_fibsom	0.0644531	0.0859375
SNU601_immune	0.0273438	0.0546875
SNU601_MM	0.0058594	0.0234375

We updated the text accordingly and included the statistical analyses:

In the section for the MM dataset: “For every other tested reference dataset, the quality of the CNV calls dropped significantly compared to the original reference from the same dataset (Wilcoxon signed rank test, FDR < 0.05; see Methods).”

In the section for the SNU601 dataset: “Here, the performance dropped only significantly for the MM dataset as reference dataset (Wilcoxon signed rank test, FDR < 0.05; see Methods).”

And in the conclusion of this section:

“Overall, we see that the choice of the reference dataset has an **less** effect on the CNV detection in aneuploid samples ~~compared to euploid samples~~. If cells from the same sample are available, they tend to be the best reference dataset. In case an external reference is necessary, biological and technical differences should be kept as small as possible. If a cell type that itself is harboring aneuploidies is used as a reference, the CNV predictions become less reliable.”

Page 14, authors mention that some methods used multithreading but mention only Numbat.

We clarified this now by adding an additional column for our method description in Table 1. Four of the six methods can apply multi-threading, InferCNV, copyKat, Numbat and SCEVAN.

We also changed the respective sentence to not highlight only one method:

“We ran each method with only one thread for better comparison, **but most of the methods offer options for multi-threading (see Table 1).**”

Page 15, It is unclear what results suggest that that loss were more difficult to identify than gains?

Thank you for this suggestion. We improved this evaluation by comparing the sensitivity to identify gains vs the sensitivity to identify losses using a Wilcoxon signed rank test (adjusted for multiple testing with Benjamini Hochberg correction). The adjusted p-values were < 0.05 for three methods (CONICSmat, Numbat (CNV) & InferCNV (CNV)) and < 0.1 for two more methods (InferCNV (Expr) & CaSpER), indicating that out of nine methods, five had more difficulty to identify losses than gains (see **New Supplementary Figure 6** below).

We clarified the text in the manuscript respectively in the result section:

“We see that **generally three** of the CNV callers (CONICSmat, Numbat (CNV) & InferCNV (CNV)) are better at predicting gains, as they show a significantly higher sensitivity for gains compared to losses (Wilcoxon signed rank test, FDR < 0.05; **Supplementary Figure 6 Figure 2A, Supplementary Figure 2**). Two more methods show at least slightly significantly higher sensitivity for gains (InferCNV (Expr) & CaSpER; FDR < 0.1).”

And in the discussion:

“~~In general~~**For several methods**, losses were more difficult to identify compared to gains.”

New Supplementary Figure 6. Comparison of sensitivity for gains vs losses for all methods across the droplet-based human datasets. “p.adj” represents the FDR-corrected p-values from Wilcoxon signed rank tests (one-sided for sensitivity for gains is greater).

Page 17: “ Bam files and count matrices were generated from the fastq files by running CellRanger (version 7.0.0), except for the BCC dataset, where the annotated count matrix was taken directly from GEO.” For the BCC dataset, you mention another Cell ranger version without giving the version.

Thank you for pointing this out. We looked up the CellRanger version that was described in the respective manuscript for the BCC dataset and added it to the manuscript:

“Bam files and count matrices were generated from the fastq files by running CellRanger (version 7.0.0), except for the BCC dataset, where the annotated count matrix was taken directly from GEO (based on CellRanger version 2.1.0).”

Reviewer #1 (Remarks on code availability):

NA.

Reviewer #2 (Remarks to the Author):

This study examines a topical and interesting area in bioinformatics - inference of copy number from single cell RNA seq data. They authors have done an extensive job in evaluating multiple callers and presented comparative metrics and made some recommendations. I think this potentially is a valuable study with revision - it is an area that this reviewer is interested in and I was not aware of the pros and cons of many of these callers. I have a couple of major suggestions for improvement.

The datasets used could be improved. They are mostly cell lines, unpaired, and even the cancer dataset apparently is compromised by the lack of paired, single cell and bulk reference WGS data. There are more comprehensive real world datasets that are publicly available - one is PMID 37216686 (extensive paired WGS with complex copy number alterations, single cell WGS and single cell RNA-seq). Others exist (e.g. medulloblastoma data from the Northcott group at St Jude). I would hesitate to move ahead with the recommendations presented here without a detailed benchmarking of optimal datasets. I would also like to see analysis of ability to resolve focal copy number alterations. That is not addressed here. Many of the visualizations refer to statistical accuracy, or genome-wide profiles, and could be improved.

Some of the callers (e.g. Numbat) appear excessively clean and may undersegment data (compared to WGS for example). This is related to the focal CNA point above.

Part 1 - more datasets

We thank the reviewer for this suggestion and added 6 new datasets: the ALL dataset suggested by the reviewer (one primary sample called iAMP21 in the following, with scRNA-seq from 10X Genomics plus scWGS), a mouse dataset (one primary sample of T-cell lymphoma with scRNA-seq from droplet-based plus scWGS) and four paired datasets, where scRNA-seq and scWGS was measured in the same cells (2 cell lines (HCT116 and A375) and 2 primary samples of acute lymphoblastic leukemia (ALL1 and ALL2), measured using DNTR-seq).

The first dataset is the primary sample ALL dataset (Gao et al. 2023), suggested by the reviewer, called iAMP21 in the following, which consists of unpaired 10X RNA-seq and scWGS data. We analyzed it using our standard pipeline and saw that the performance results matched generally well with our previous findings, its performance being very close to the mean performance across datasets (see Figure below).

We also included this dataset in the section where we assess how well the methods automatically detect cancer cells. Here, copyKat and SCEVAN did not perform well on the iAMP21 dataset. The bad performance of copyKat is in line with our previous findings, namely that copyKat has issues when very few non-tumor cells are present in the dataset. For SCEVAN, further exploration showed that in the initial step of assigning confidently normal cells based on marker genes, SCEVAN wrongly assigned 7 leukemia cells as B-cells, T-cells or macrophages, respectively. We conclude from this that the marker genes used in SCEVAN might not work as well in each cancer type. We included the iAMP21 results into the respective sections of “Benchmarking single-cell RNA-seq CNV predictions compared to genomic ground truth in droplet-based data” and “Benchmarking automatic identification of tumor cells”.

Revision Figure. Performance of the new datasets compared to the previously added datasets (called “old_datasets”). Error bar shows the standard deviation of the previously used datasets.

To increase the number of primary samples, as suggested by the reviewer, we included additionally a new mouse dataset, a T-cell lymphoma, with unpaired scWGS and scRNA-seq data from SeekGene. This gives us also insights into CNV calling in other species than humans. The mouse data performed quite well for all methods that rely only on the expression matrix, with performance results within and above the standard deviation of the previous results. Highlighting complications related to the methods that use allelic frequencies as input, we could not run Numbat on this dataset, as we had no haplotype information available for this mouse species. We ran CaSpER, but the tool did not identify any SNPs and so we could not estimate CNVs with it.

Based on these results, we can extend the conclusions from the previous performance results to other species, in this case mice, for the expression based methods. Interestingly, this analysis of mouse data shows that for allele-frequency based methods, not all mouse strains might show enough heterozygosity and the haplotype profile needs to be known to be able to provide CNV results, as discussed now in the manuscript.

Following the suggestion of the reviewer, we furthermore included paired datasets, where the WGS and RNA were measured in the same cells. For this, we included two cell lines, HCT116 and A375, and two primary cancer samples, ALL1 and ALL2, measured with DNTR-seq (Zachariadis et al. 2020). To our knowledge, all the currently existing paired single cell WGS-RNA are plate-based, with a limited number of cells and an RNA library preparation method similar to Smart-seq2. Despite the fact that some CNV callers like copyKat, SCEVAN and Numbat were specifically developed for droplet-based methods, all

CNV methods provide analysis suggestions for Smart-seq2 data, which we added to our pipeline.

We see that the pseudobulk level performance of the expression-only CNV callers are very similar for HCT116 and A375 compared to the previous datasets. However, the two allele-frequency based methods, CaSpER and Numbat did not perform well. For the HCT116, Numbat found no CNVs at all and CaSpER performed worse than the other methods. For the A375, CaSpER found no CNVs at all and Numbat performed badly.

A potential reason for the decreased performance is that the two associated SNP callers of CaSpER and Numbat identified far fewer SNPs in both datasets compared to the previous droplet-based datasets (see **New Supplementary Figure 17**). This is most likely caused by the small number of cells and is relevant for both methods, as they do the SNP calling on pseudobulk level (CaSpER) and subclone level (Numbat), respectively.

Supplementary Figure 17. Number of SNPs identified with CaSER and Numbat (axes in log-scale) for the two DNTR-seq based methods A375 and HCT116 compared to the human droplet-based datasets analyzed in this benchmarking.

For the ALL datasets, no genotype information was available, and therefore, we were only able to run the expression-only CNV callers. Both ALL1 and ALL2 showed good performance across all expression-only methods.

The paired data allows us to do a proper cell-by-cell comparison additionally to the standard pseudobulk comparison performed previously (see **New Supplementary Figure 20** below). In general, the per cell results were lower than the pseudobulk ones, in line with what we had observed for the unpaired datasets. Additionally, the use of truly paired data allowed us to study how some clones could be recovered more faithfully than others.

We introduced a new results section with these results, as both the mouse dataset and the paired datasets present different characteristics than the unpaired human data. This section, called “Benchmarking CNV prediction in other organisms and technologies”, is the following:

“We extended the performance evaluation using other single cell technologies and organisms. This includes paired methods, which enable the analysis of RNA and WGS in the same cell and thereby a per cell comparison of the CNV prediction results between RNA and WGS. In total, we included four paired plate-based datasets, measured with DNTR-seq²²,

and one mouse dataset (droplet-based) (**Supplementary Table 1**). In principle, all methods allow to run data generated by plate-based RNA technologies and data from other species apart from human. The newer CNV callers copyKat, Numbat and SCEVAN, however, were developed mainly for human data generated by droplet-based methods.

First, we analysed the CNV results for the two paired plate-based cell lines HCT116 and A375. For the expression-based methods, the performance was close to the mean performance of the droplet-based datasets (**Figure 2A, Supplementary Figure 4**). However, for the methods including AF information, CaSpER and Numbat, the performance dropped considerably, with few to no CNVs identified (**Supplementary Figure 15+16**). This could be due to the fact that, compared to droplet-based datasets, an order of magnitude fewer SNPs could be called, likely due to the low number of cells per experiment (**Supplementary Figure 17**). We analysed the other two plate-based paired datasets, ALL1 and ALL2, only using the expression-based methods, where they showed above average performance (**Figure 2A, Supplementary Figure 4**).

Thanks to the paired information, we could perform a cell by cell comparison of CNV profiles (**Supplementary Figures 16-20**). The HCT116 cell line and the ALL2 dataset were very homogenous based on the DNA CNVs, and the same homogeneity was visible at the RNA CNV level (**Supplementary Figures 15+19**). In contrast, the A375 cell line and the ALL1 dataset showed more heterogeneous CNV profiles across cells in the DNA data (**Supplementary Figures 16+18**). For example, the ALL1 dataset contained two subclones, the larger with an additional gain in chromosome 6. All methods performed significantly better at predicting the profiles of the larger subclone (Wilcoxon rank sum test, FDR < 0.05; **Supplementary Figure 20C**). Generally, the per cell performance was on average lower than the pseudobulk performance (**Supplementary Figure 20**), similar to the results for the SNU601 cell line discussed previously (**Supplementary Figure 14**). Especially for ALL2, it was visible again that the methods which groups cells into subclones (InferCNV (CNV) and SCEVAN (CNV)) had estimates which were closer to the pseudobulk results.

Finally, we tested the performance of the methods on another species, a mouse droplet-based dataset from a T-cell lymphoma. Some of the CNV RNA methods are restricted by the organisms they can run on. In particular copyKat, Numbat and SCEVAN offer only human and mouse annotations as options. All expression-based methods showed very good performance on the tested mouse data, above average compared to the human droplet dataset (**Figure 2A, Supplementary Figure 4**). Numbat requires haplotype information, which was not available for the analysed mouse dataset and therefore could not be tested here. CaSpER could not detect SNPs on this dataset.

In conclusion, we saw that the CNV prediction methods also worked for scRNA-seq plate-based technologies and non-human species. However, the methods which incorporate AF information were more restricted due to the low number of inferred SNPs in the plate-based datasets and the need of haplotype information. These methods performed worse than the expression-based ones for the plate-based datasets.”

New Supplementary Figure 20. Per cell performance results (Pearson correlation) with matched cells for the four DNTR-seq datasets: HCT116 (**A**), A375 (**B**), ALL1 splitted after the two subclones (**C**) and ALL2 (**D**). The vertical black line visualizes the correlation to the pseudobulk. For the HCT116 (**A**), Numbat (CNV) is missing here, as no CNVs were found for this dataset, so no correlation could be calculated. For the same reason, CaSpER is missing for the A375 (**B**) (no CNVs found with this method). For ALL1 and ALL2 (**C+D**), only the expression-based methods were tested.

We want to point out that paired scRNAseq and scWGS datasets are not very well established, with only few existing datasets compared to the large number of single-omics RNA-seq datasets. This is also the reason for the large interest in the CNV RNA callers used in our study, as corresponding scWGS data is difficult to generate and only for a small number of cells. Additionally, most published datasets are based on the 10X Genomics technology (or similar droplet-based methods), while the paired datasets are all plate-based to the best of our knowledge. As we showed, the performance differs between droplet- and plate-based and it is important to consider and evaluate both.

Part 2 - differences between cell lines and primary cancer samples

We thank the reviewer for mentioning potential differences between cell lines and primary cancer samples. To evaluate this, we tested now whether there are statistically differences in the method performance between cell lines and primary cancer samples across our 20 datasets. Using the Wilcoxon Rank Sum test (two-sided), we found no statistical differences in F1 scores between cell lines and primary samples for any of the methods (FDR > 0.05 in all cases; see also Revision Figure below):

method	p	fdr
InferCNV (CNV)	1.0000000	1.0000000
InferCNV (Expr)	0.8167957	1.0000000
CaSpER	0.5208791	1.0000000
copyKat	0.5880289	1.0000000
SCEVAN (CNV)	0.9384675	1.0000000
SCEVAN (Expr)	0.5355521	1.0000000
Numbat (Expr)	0.0790703	0.7116331
Numbat (CNV)	0.7703297	1.0000000
CONICSmat	0.2749226	1.0000000

We added this additional test to our discussion:

“We included in the benchmarking both primary cancer samples and cell lines, which differ slightly in their analysis, as cell lines require external reference datasets. This issue applies not only to the commercial cell lines we used in our test cases, but also to any newly established cell line in the laboratory. This makes our results, specifically for analyzing cell lines, relevant for different scenarios. **We see, however, no significant differences in method performance between the cell lines and the primary cancer samples (Wilcoxon rank sum test, FDR > 0.05).**”

Our observation that the CNV calling methods perform very similarly on cell lines and primary cancers is also reasonable from a biological point of view. CNVs are a basic genomic alteration, that lead to the same signal at the single-cell level in both cell lines and primary samples, compared to more complex biological processes which might not be faithfully reflected in cell lines. With this in mind, using cell lines for testing CNV calling methods is a suitable strategy.

Revision Figure. Performance differences between all tested cell lines and primary cancer datasets.

Part 3 - focal CNVs

We thank the reviewer for the suggestion to explore also focal CNVs. In our analyses so far, we did not distinguish the CNV sizes by length, but we agree that this is an interesting extension. Unfortunately, the definition of focal CNV is not unique, therefore we decided to explore two different definitions:

1. Focal type 1: Define a focal CNV as smaller or equal than 3Mb, following these publications (Bierkens et al. 2013; Krijgsman et al. 2014; Pös et al. 2021)
2. Focal type 2 & 3: Define a focal CNV to be smaller than the chromosome arm, following these publications (Beroukhim et al. 2010; Zack et al. 2013). For this, we define that a focal CNVs needs to lie completely inside the chromosome arm and the length needs to be <50% or < 95% of the length of the chromosome arm (Focal type 2 or 3, respectively).

For our datasets with scWGS ground truth, we use Aneupfinder for identification of CNVs, which has a resolution of 100kB, so that focal CNVs are identifiable in these datasets. We group adjacent bins having the same CNV state together and check the length distribution of the results, here exemplarily shown for the SNU601 dataset (the horizontal bar represents the 3Mb cutoff):

The focal type 2 and 3 definitions are defined based on the length of the chromosome arm, as shown here again exemplarily for the SNU601 dataset. Focal type 2 is smaller than 50% of a chromosome arm and focal type 3 smaller than 90% of a chromosome arm:

New Supplementary Figure 7. Method performance on focal CNVs. A. Number of total and focal CNVs identified from scWGS based on three different focal definitions: focal type 1 (size < 3Mb), focal type 2 (size < 50% of a chromosome arm) and focal type 3 (size < 95% of a chromosome arm). B. Percentage of CNVs that overlap the analysed regions of the scRNA-seq CNV callers for focal type 1 CNVs and broad CNVs (all CNVs > 3Mb). C. Sensitivity in identifying gains and losses for each of the focal types.

We identified several focal CNVs from the scWGS ground truth (according to the different definitions, see **New Supplementary Figure 7A**). However, we saw that scRNA-seq data is not the best tool for identifying very small CNVs (following focal definition 1), due to the sparse genomic coverage in this data type. When estimating what percentage of the focal type 1 and broad CNVs (CNVs > 3Mb) are overlapping with the scRNA-seq data, far more broad than focal CNVs could be identified (**New Supplementary Figure 7B**).

We explored next how well the different tools could identify the focal CNVs, choosing the F1 cutoffs as estimated from the previous analyses. We again calculated the sensitivity, specifically for the focal CNVs, using the optimal F1 scores as estimated in the manuscript. We see that all methods have a lower sensitivity for detected CNVs of focal type 1 compared to the general sensitivity of detection CNVs (**New Supplementary Figure 7C**). Only CaSpER gets at least a similar sensitivity for focal type 1 CNVs and all CNVs for the gains. In contrast, for all methods the sensitivity was similarly high for focal type 2 and 3 compared to all CNVs (even higher in some of the cases). The reviewer had the hypothesis that Numbat, which showed less segments, might for this reason potentially be worse at identifying focal CNVs, but we could not prove this connection.

We added a section about focal CNVs into our manuscript to describe these results, together with the New Supplementary Figure 7 shown above.

Copy number variations can be further classified, based on their length, into focal CNVs of short length and broad CNVs²². There are different biological mechanisms behind both types, and often focal CNVs are associated with cancer-driving genes²³. However, the detection of focal CNVs is difficult due to their size, prompting us to analyze specifically how well the different callers can identify them. The high resolution of the ground truth scWGS data allowed us to categorize the CNVs into focal and broad for the respective datasets. We tested three different definitions of focal amplifications, based on their size: focal type 1 (size < 3Mb)^{1,23,24}, focal type 2 and focal type 3 (size < 50% or < 95% of a chromosome arm, respectively)^{22,25}. All datasets contained several focal CNVs for all three definitions, however for focal type 1 CNVs, only a few were overlapping the regions analyzed by the CNV calling methods (**Supplementary Figure 7A+B**). This highlights a general coverage problem of all scRNA-seq callers for very small CNVs. Also the sensitivity of the individual callers for this subset of focal type 1 CNVs was very low (**Supplementary Figure 7C**). However, the detection sensitivity of focal type 2 & 3 CNVs was similarly high to the one for all CNVs, so we see that only focal CNVs according to the most strict definition are affected. For this subset, we conclude that all scRNA-seq callers are in general not suitable for identifying them.

And a respective section in the methods:

We analyzed the performance to identify focal CNVs for all human droplet-based datasets, for which a scWGS ground truth dataset was available. As before, the CNV calls from the scWGS were first discretized: loss values are defined by at least 50% of the cells having a loss at this segment and gain values as at least 50% of cells having a gain there. Next, neighbouring segments with the same CNV status were combined. All CNVs (gains or losses) smaller than 3MB were defined as focal type 1. Next, the overlap between CNVs and chromosome arms was calculated, CNVs that lie completely within a chromosome arm and have additionally a size smaller than 50% or 95% of the overlapping chromosome arm, respectively, were defined as focal CNVs type 2 and 3. Then, the sensitivity of detecting gains and losses for each of the focal classes was calculated with the same approach as for the general CNVs, taking the same optimal gain and loss thresholds estimated before for the general CNVs.

Part 4 - visualization

We are happy to update our visualisations, but we would appreciate more guidance and details on what the reviewer feels could be improved.

*Reviewer #2 (Remarks on code availability):
I do not have time.*

References

- Beroukhim, Rameen, Craig H. Mermel, Dale Porter, Guo Wei, Soumya Raychaudhuri, Jerry Donovan, Jordi Barretina, et al. 2010. "The Landscape of Somatic Copy-Number Alteration across Human Cancers." *Nature* 463 (7283): 899–905. <https://doi.org/10.1038/nature08822>.
- Bierkens, Mariska, Oscar Krijgsman, Saskia M. Wilting, Leontien Bosch, Annelieke Jaspers, Gerrit A. Meijer, Chris J. L. M. Meijer, Peter J. F. Snijders, Bauke Ylstra, and Renske D. M. Steenbergen. 2013. "Focal Aberrations Indicate *EYA2* and *hsa-miR-375* as Oncogene and Tumor Suppressor in Cervical Carcinogenesis." *Genes, Chromosomes and Cancer* 52 (1): 56–68. <https://doi.org/10.1002/gcc.22006>.
- Gao, Qingsong, Sarra L. Ryan, Ilaria Iacobucci, Pankaj S. Ghate, Ruth E. Cranston, Claire Schwab, Abdelrahman H. Elsayed, et al. 2023. "The Genomic Landscape of Acute Lymphoblastic Leukemia with Intrachromosomal Amplification of Chromosome 21." *Blood* 142 (8): 711–23. <https://doi.org/10.1182/blood.2022019094>.
- Kinker, Gabriela S., Alissa C. Greenwald, Rotem Tal, Zhanna Orlova, Michael S. Cuoco, James M. McFarland, Allison Warren, et al. 2020. "Pan-Cancer Single-Cell RNA-Seq Identifies Recurring Programs of Cellular Heterogeneity." *Nature Genetics* 52 (11): 1208–18. <https://doi.org/10.1038/s41588-020-00726-6>.
- Krijgsman, Oscar, Beatriz Carvalho, Gerrit A. Meijer, Renske D.M. Steenbergen, and Bauke Ylstra. 2014. "Focal Chromosomal Copy Number Aberrations in Cancer—Needles in a Genome Haystack." *Biochimica et Biophysica Acta (BBA) - Molecular Cell Research* 1843 (11): 2698–2704. <https://doi.org/10.1016/j.bbamcr.2014.08.001>.
- Pös, Ondrej, Jan Radvanszky, Gergely Buglyó, Zuzana Pös, Diana Rusnakova, Bálint Nagy, and Tomas Szemes. 2021. "DNA Copy Number Variation: Main Characteristics, Evolutionary Significance, and Pathological Aspects." *Biomedical Journal* 44 (5): 548–59. <https://doi.org/10.1016/j.bj.2021.02.003>.
- Zachariadis, Vasilios, Huaitao Cheng, Nathanael Andrews, and Martin Enge. 2020. "A Highly Scalable Method for Joint Whole-Genome Sequencing and Gene-Expression Profiling of Single Cells." *Molecular Cell* 80 (3): 541-553.e5. <https://doi.org/10.1016/j.molcel.2020.09.025>.
- Zack, Travis I, Steven E Schumacher, Scott L Carter, Andrew D Cherniack, Gordon Saksena, Barbara Tabak, Michael S Lawrence, et al. 2013. "Pan-Cancer Patterns of Somatic Copy Number Alteration." *Nature Genetics* 45 (10): 1134–40. <https://doi.org/10.1038/ng.2760>.

Point-by-point response to the reviewers

We would like to thank both reviewers again for their valuable comments and for having taken the time to review our work a second time. We appreciate that both reviewers see the improvements of our manuscript and provide in the following a point-by-point answer to the last open comments.

Reviewer comments are marked in green and italic font.

Responses are typeset in regular font.

Edits in the main text are marked in brown.

Reviewer #1 (Remarks to the Author):

Authors have carefully responded to most of our comments.

We have reviewed the Github repository (https://github.com/colomemaria/benchmark_scrnaseq_cnv_callers).

The README is clear and provides detailed information. However, we have not attempted to implement these pipelines.

We urge the authors to remove the truncated AUC method from the manuscript. This method is not intuitive and may not be well-received by the community. AUC has been around forever, and it is never truncated.

If some values are inappropriate to predict a deletion or a duplication, they should not be integrated into the predictive model. If aberrant values can be clearly defined beforehand, they can also be removed from the predictive model.

We thank the reviewer for their second round of feedback and are glad that they appreciate the improved manuscript and our github page.

Thanks to the reviewer's comment we realized that our concept of truncated AUC is still not well enough introduced in the manuscript. The metric is based on the idea of partial AUCs, which is well established in the field (McClish 1989; Walter 2005; Carrington et al. 2020; Chaibub Neto et al. 2024). A partial AUC allows the calculation of the area under the curve for a specific range based on sensitivity and/or specificity thresholds and has different applications. The general motivation is that only a certain area of the curve is of interest for the analysis.

In our application, we restrict the AUC based on a maximal sensitivity threshold. As discussed, each method has a certain baseline value and only values higher than it are reasonable for the gain threshold and values lower for the loss threshold. For example in the method copyKat, the baseline is fixed at 0 and a region can only be called a gain if the value is >0 , or a loss if the value is <0 . This means that for every method, the sensitivity can never become higher than the sensitivity at this minimal gain threshold or maximal loss threshold, respectively. For copyKat and the MCF7 dataset (the example depicted in Supplementary Figure 2 in the manuscript), the maximal reachable sensitivity for gains is 62%, when considering all values larger than 0 as gain. This is shown in the figure by the vertical line. Therefore, we calculate the partial AUCs until this value. For the implementation, we use the R package pROC, which has within its `auc()` function the parameter "partial.auc" for these cases.

Supplementary Figure 2. Illustration of the **truncated partial** AUC scores, exemplarily shown here for the ROC curve for CopyKat, for identified gains applied to the MCF7 dataset. The area under the curve (AUC) is only calculated for thresholds which are inside the allowed value range for the CNV type and method. Here, for gains the only allowed thresholds are above 0 (for losses the allowed thresholds would be below 0). The vertical line visualizes the sensitivity when using 0 as threshold.

The suggestion of the reviewer to remove aberrant values beforehand would not have the same effect. If the values are removed completely beforehand, then they are not considered at all downstream when calculating the AUC. Therefore, with this suggestion, the presence of aberrant values would not penalize the methods that produce them. Instead, the methods that produce many aberrant values should be penalised, with a lower AUC value given by the partial AUC score. On top, the suggestion of the reviewer would mean removing different values for different methods, so that a fair comparison between methods is not given anymore.

We acknowledge that the term “partial AUC” is much more widely used in the community than the alternative term “truncated AUC” and decided to update everywhere in the manuscript the terminology to partial AUC.

We additionally updated the manuscript text to refer to previous uses of partial AUC to clarify this further:

“We applied threshold-independent evaluation metrics using the correlation and AUC scores. For the AUC scores, predictions were evaluated separately for gain versus all and loss versus all, resulting in two scores. Not the complete range of thresholds is biologically meaningful to classify regions as gains or losses, respectively, as every method defines a baseline score. For this reason, we chose to **additionally calculate a partial AUC^{21,22}, with a maximal sensitivity defined by the baseline score implement a truncated version of the AUC scores where so that** only thresholds up to the baseline score were evaluated for losses and only scores higher than the baseline score were evaluated for gains (see Methods, Supplementary Figure 2). Of note, **truncated partial** AUC values below 0.5 indicate that most thresholds are outside the biologically meaningful value range (see Methods).”

And in the methods:

However, not all thresholds are reasonable for gain or loss prediction, respectively. Every method has a baseline value, usually 0, and only loss values lower than the baseline and gain values higher than the baseline are biologically reasonable. Therefore, we calculated additionally a **partial AUC restricted for sensitivities between 0 and s_max. s_max is thereby defined by the baseline**. We reported the **truncated partial AUC** loss and gain values on top of the standard AUC values.

Reviewer #2 (Remarks to the Author):

Overall, the authors have been highly responsive to the suggestions of this reviewer, including analysis and benchmarking of callers across multiple additional datasets. This was the main area for improvement.

A couple of extra points if the manuscript proceeds:

Be careful about designating "copy number" subclones based on subchromosomal changes on chromosome 6, particularly 6p (the part of the rebuttal and manuscript that refers to SFig 20). This can be a pseudo CN change due to coordinate change in expression of histone genes due to a subpopulation of cells being in cell cycle - not a real copy number based subclone.

My suggestion about improving visualization referred to the fact that the number of figures in the main MS is not that great, and many refer to benchmarking/statistical analyses rather than depictions of actual regions/samples/data to get a good feeling for real world performance of the callers.

We thank the reviewer for their second round of feedback and are glad that they appreciate the improved manuscript. With regards to the last two comments:

1. The subclonal definition of the ALL1 dataset

Thanks for pointing this out, this was not explained clearly enough in the manuscript. We derived the subclone definition directly from the original authors of the publication, which categorized the cells based on the scWGS. This means, the cells show differences in DNA levels, this classification can not be influenced by changes in RNA levels, i.e. expression changes of histones. We updated the manuscript to make this clearer:

~~For example~~Based on CNVs called from the scWGS data, the cells of the ALL1 dataset contained two subclones were categorized into two subclones in the original publication²⁸, the larger with an additional gain in chromosome 6."

Additionally, we checked whether the cells from the two subclones showed differences in their cell cycle, using the standard workflow from Scanpy for cell cycle assignment based on marker genes. This lead us to this annotation of cells:

	diploid cells	clone 1	clone 2
G1 phase	48 cells	71 cells	8 cells
G2M phase	18 cells	41 cells	8 cells

S phase	19 cells	42 cells	7 cells
---------	----------	----------	---------

The cells from both clones as well as diploid cells were distributed across all three cell cycle phases. There is therefore no clear trend pointing at the fact that the clonal differences are driven by the cell cycle state.

Nevertheless, we agree that this is a relevant aspect for the benchmarking. Although this was not the case for our analysis, we still added a sentence about it to our discussion:

“Finally, all methods provide information about the CNV clones present in the analyzed sample. CopyKat, InferCNV and Numbat were able to identify the right clonality, while on the opposite side of the spectrum SCEVAN was not able to disentangle any clones. **In general, CNV clones called from scRNA-seq data could also be caused by regulatory changes of gene clusters that lead to expression changes, such as cell cycle effects, and need to be evaluated carefully.**”

2. Visualizations

The reviewer is correct that most visualizations currently represent the summary statistics. As one of our main outcomes was that the performance of the callers depends a lot on the dataset, showing specific datasets / regions alone can be misleading as it is not possible to deduce general recommendation from individual datasets.

For showing real data, we always display the karyograms showing the results of the CNV callers, never the raw scRNA-seq count matrices. The scRNA-seq raw data is extremely noisy and no clear patterns are visible, here exemplarily shown for 100 cells of the SNU601 dataset:

Only strict filtering and additional normalization makes the CNV patterns visible within the RNA data, which is a crucial part of each CNV RNA caller and results in the karyograms we display.

Nevertheless, we agree that more karyograms in the main text could give the reader a better feeling for the analyses. We added a second karyogram for Figure 2E depicting the A375 dataset to show also one plate-based dataset in the main manuscript. For Figure 3A+B, we added two karyograms showing how the performance differs for the diploid dataset depending on the reference dataset, which were before in the supplement (Supplementary Figure 21). We also created additional karyograms to visualize how the performance

changes, whether the methods apply their own automatic cell type annotation or use our manual cell type annotations. We put one karyogram for iAMP21 in Figure 3F and the ones for the remaining datasets in the new Supplementary Figure 24. The updated figures are all shown below and we made sure to have all detailed visualizations in the supplement.

Figure 2. General performance evaluation on aneuploid datasets. (A) Performance comparison across all datasets. Due to lack of genomic information, the ALL1, ALL2 and mouse data was not run with CaSpER and Numbat. For the dataset A375, CaSpER

identified no CNVs, for HCT116, Numbat (CNV) identified no CNVs. (B) Impact of dataset characteristics on the performance (maximal F1 scores). The features here show the total number of UMI counts, the number of cells and the number of expressed genes in the cancer dataset, the mean dropout rate per cell, the mean coefficient of variation across genes, the fraction of gained regions from all ground truth annotated regions and the fraction of CNV regions (gained and lost together). These are the top 7 general dataset characteristics with the highest mean absolute correlation to the performance, all features are shown in the supplement (**Supplementary Figure 8**). (C) Method comparison within the SNU601 dataset. (D+E) Karyogram of the SNU601 dataset (D) and the A375 dataset (E), every method score was scaled to have the same standard deviation.

Figure 3. A+B. Karyograms of CNVs in CD4+ T cells when using either CD4+ T cells (A) and CD14+ Monocytes (B) from the same dataset as reference cells. **C.** Root mean squared error (RMSE) between CNV predictions of each method and diploid baseline as ground truth. The methods with lowest RMSE perform best. Panel title shows chosen reference cells. **D+E.** Performance of the methods with different reference datasets for the MM dataset (D) and the SNU601 dataset (E). For the MM dataset, we tested a second healthy PBMC dataset, which we split into B cells, T cells and Monocytes (mono). We additionally tested a

healthy gastric dataset and a gastric cancer dataset (SNU601). For the SNU601, we tested a second healthy gastric reference, which we split into three groups, epithelial and endothelial cells (epiendo), fibroblasts and smooth muscle cells (fibsom) and immune cells. We also tested the MM dataset as reference. **F. Karyograms showing the difference between using a manual reference and an automatic one for the dataset iAMP21.**

Reviewer #3 (Remarks to the Author):

References

- Carrington, André M., Paul W. Fieguth, Hammad Qazi, Andreas Holzinger, Helen H. Chen, Franz Mayr, and Douglas G. Manuel. 2020. "A New Concordant Partial AUC and Partial c Statistic for Imbalanced Data in the Evaluation of Machine Learning Algorithms." *BMC Medical Informatics and Decision Making* 20 (1): 4. <https://doi.org/10.1186/s12911-019-1014-6>.
- Chaibub Neto, Elias, Vijay Yadav, Solveig K. Sieberts, and Larsson Omberg. 2024. "A Novel Estimator for the Two-Way Partial AUC." *BMC Medical Informatics and Decision Making* 24 (1): 57. <https://doi.org/10.1186/s12911-023-02382-2>.
- McClish, Donna Katzman. 1989. "Analyzing a Portion of the ROC Curve." *Medical Decision Making* 9 (3): 190–95. <https://doi.org/10.1177/0272989X8900900307>.
- Walter, S. D. 2005. "The Partial Area under the Summary ROC Curve." *Statistics in Medicine* 24 (13): 2025–40. <https://doi.org/10.1002/sim.2103>.

Point-by-point response to the reviewers

We would like to thank all three reviewers for their time reviewing our manuscript and their positive response to our last version.

Reviewer comments are marked in green and italic font.

Responses are typeset in regular font.

Reviewer #1 (Remarks to the Author):

Authors have responded to our comments. They improved the description and explanation of the partial AUC method. However, this approach will make comparing AUC across studies more difficult (Chaibub Neto et al. 2024. "A Novel Estimator for the Two-Way Partial AUC." BMC Medical Informatics and Decision Making 24 (1): 57. <https://doi.org/10.1186/s12911-023-02382-2>).

Thanks for the positive response. We agree that the partial AUC is more difficult to interpret than the standard AUC. However, given our explanations in the last response, we still believe this is the best solution to deal with the restricted value range.

Reviewer #2 (Remarks to the Author):

The authors have addressed my comments

Reviewer #3 (Remarks to the Author):
